# Effects of horizontal resolution and air-sea coupling on simulated moisture source for East Asian precipitation in MetUM GA6.0/GC2.0

Liang Guo[1], Ruud J. van der Ent[2], Nicholas P. Klingaman[1], Marie-Estelle Demory[3], Pier Luigi Vidale[1], Andrew G. Turner[1,4], Claudia C. Stephan[5], and Amulya Chevuturi[1]

[1]National Centre for Atmospheric Science, Department of Meteorology, University of Reading, United Kingdom
[2]Department of Water Management, Faculty of Civil Engineering and Geosciences, Delft University of Technology, Delft, the Netherlands
[3]Department of Environmental Systems Science, Institute for Atmospheric and Climate Science, ETH Zürich, Zürich, Switzerland
[4]Department of Meteorology, University of Reading, Reading, United Kingdom
[5]Max Planck Institute for Meteorology, Hamburg, Germany

**Correspondence:** Liang Guo (l.guo@reading.ac.uk)

**Abstract.** Precipitation over East Asia in six MetUM simulations are compared with observation and ERA-Interim reanalysis. These simulations include three different horizontal resolutions, from low, medium to high, and including atmosphere-only version (GA6.0) and air-sea coupling version (GC2.0). Precipitations in simulations are systematically different from that of observation and reanalysis. Increasing horizontal resolution and including air-sea coupling improve simulated precipitation but cannot eliminate bias. Moisture sources of East Asian precipitations are identified using the WAM-2layers - a moisture tracking model that traces moisture source using collective information of evaporation, atmospheric moisture and circulation. Similar to precipitation, moisture sources in simulations are systematically different from that of ERA-Interim. Major differences in moisture sources include underestimated moisture contribution from tropical Indian Ocean and overestimate contribution from Eurasian continent. By increasing horizontal resolution, precipitation bias over the Tibetan Plateau is improved. From the moisture source point of view, this is achieved by reducing contribution from remote moisture source and enhancing local contribution over its eastern part. Although including air-sea coupling does not necessarily change East Asian precipitation, moisture sources show differences between coupled and atmospheric-only simulations. These differences in moisture sources indicate different types of models biases caused by surface flux or/and atmospheric circulation on different locations. These information can be used to target model biases on specified locations and due to different mechanisms.

*Copyright statement.* TEXT

# 1 Introduction

Identifying moisture source for East Asian (EA) precipitation has been a challenging subject that motivates the scientific community and is essential for regional socio-economical development. Different methods have been applied from the diagnosis of the net moisture flux on the boundary of a studied region (e.g., Zhou and Yu, 2005), to using one or two-dimensional analytic models (e.g., Guo et al., 2018), to using moisture tracking models based on the atmospheric moisture conservation under the both Lagrangian and Eulerian frameworks (e.g., Wei et al., 2012; Zhang et al., 2017; Guo et al., 2019; Fremme and Sodemann, 2019). The understandings about the major moisture source for EA precipitation are changing. As East Asia being under the influence of the East Asian summer monsoon, early studies tend to consider the adjacent oceans being the major direct moisture source for the EA precipitation and its interannual variability (Zhou and Yu, 2005; Wang and Chen, 2012). With sophisticated moisture tracking tools being applied, moisture source for precipitation can be identified more accurately. As a result, moisture contributions of land surface have been recognised (Wei et al., 2012; Zhang et al., 2017; Fremme and Sodemann, 2019) and replace oceanic sources becoming major contributors to EA precipitation, especially over the northern and western parts. Considering the fact that EA spans a large area from tropics to extra-tropics and experiences dry winter and wet summer, the major land moisture source also varies from region to region and from season to season. In summer over southeastern EA, take the Yangtze River (YR) region for example, the major moisture source is the adjacent land along the route of the EA summer monsoon, i.e., the southwestern China and the Indo-China peninsula (Zhao et al., 2016; Fremme and Sodemann, 2019). One the other hand, in winter over mid-latitude EA, the major moisture source is the vast Eurasian continent beneath the mid-latitude westerly jet (van der Ent et al., 2010). On Tibetan Plateau, due to the surrounding mountains, the major moisture source is the evaporation from the local land mass (Curio et al., 2015; van der Ent and Savennije, 2011; Zhang et al., 2017). Although more and more recent studies support the view that the land surface is the major moisture source for EA precipitation, exceptions have also been found. Guo et al. (2019) shown that, during the boreal winter, due to the frozen Eurasian continent and snow cover, the mid-latitude ocean again become the major moisture source for mid-latitude EA precipitation.

Correctly simulating the global hydrological cycle in up-to-date Global Climate Models (GCMs) remains challenging (Liepert and Previdi, 2012). This challenge also remains on the regional scales and has been reported over EA (Wen et al., 2016; Yang et al., 2014; Ou et al., 2013; Chen and Sun, 2015; Jiang et al., 2015). Albeit these uncertainties, improvements in precipitation as well as on hydrological cycle have been made in GCMs with the increase of horizontal resolution and the inclusion of air-sea coupling. By investigating eighteen GCMs with horizontal resolution varying between 100 km and 20 km, Vannière et al. (2018) found improvement in precipitation over land with the increase of horizontal resolution. They also found improvement in precipitation pattern and amplitude over regional scale due to improvement in the seasonal circulation with the increase of horizontal resolution. Similar improvement in the global hydrological cycle has also been reported (Terai et al., 2018; Demory et al., 2014). Improvements in the regional scales due to increasing horizontal resolution have been reported over South Asia (Johnson et al., 2016; Ogata et al., 2017), Maritime Continent (Schiemann et al., 2014), tropical Africa (Vellinga et al., 2016) and mid-latitude storm track (van Haren et al., 2015). Over EA, Stephan et al. (2018) found that the seasonal

mean precipitation and its interannual variability are improved with increasing resolution in the Met Office Unified Models, particularly near orography in southwest China.

The ocean plays an important role in the global hydrological cycle as about 85 per cent of the evaporation and 77 per cent of the precipitation occur over the ocean (Schanze et al., 2010). The air-sea coupling makes the air-sea fluxes in the GCMs more realistic, in terms of both heat and water, therefore, changes water/precipitation distributions in models (Ratnam et al., 2015; Dong et al., 2017; Hirons et al., 2018). Furthermore, air-sea coupling changes the atmospheric circulation and the atmospheric internal variability (Barsugli and Battisti, 1998; Dickinson, 2000; He et al., 2017; Park et al., 2005; Ma et al., 2015), which changes the moisture transport and the associated precipitation over land.

Aforementioned studies show progressive understandings on the EA precipitation moisture source, and show evidences and challenges on improving the simulated hydrological cycle and regional precipitation in terms of changing the horizontal resolution and introducing air-sea coupling. Based on these knowledges, we will try to understand systematic errors in EA precipitation simulated from a set of GCMs by linking these errors to errors in evaporation and moisture transports using a moisture tracking model. The set of GCMs is the Met Office Unified Model (MetUM) Global Atmosphere 6.0 (GA6) and Global Coupled model 2.0 (GC2) with three different horizontal resolution configurations. The moisture tracking model is the Water Accounting Model (WAM-2layers). WAM-2layers has been applied to EA precipitation in previous studies (Keys et al., 2014; Guo et al., 2019) with different reanalysis datasets. Compared to other tracking methods, its efficiency makes it better tool to work with high-resolution and long-term climate simulations. More details about MetUM and WAM-2Layers are given in Section 2. Simulated precipitation and moisture sources are compared to observation and reanalyses in Section 3. Differences of the moisture source for EA precipitation due to changes in horizontal resolution and air-sea coupling are discussed in Section 4. Conclusion and discussions will be in Section 5 and 6.

## 2 Data and Methods

### 2.1 Data

The European Centre for Medium-Range Weather Forecasts interim reanalysis data set (ERA-Interim; Berrisford et al., 2011; Dee et al., 2011) is used to validate simulated precipitation and to drive the WAM-2layers moisture tracking model. Daily mean variables on single level (precipitation, evaporation, surface pressure and near-surface specific humidity) and model levels (horizontal wind and specific humidity) are used. ERA-Interim data with the horizonal resolution of $1.5° \times 1.5°$ is used to drive the WAM-2layers. This resolution is close to the model resolution at its lowest configurations (see below).

Observational daily precipitation over Asian monsoon region is obtained from the Asian Precipitation-Highly-Resolved Observational Data Integration Towards Evaluation (APHRODITE; Yatagai et al., 2012) dataset. APHRODITE utilises rain-gauge data with processes of quality control and is available from 1951-2015. To match with MetUM simulations, the period between 1982-2012 is used for both ERA-Interim and DPHRODITE. Other precipitation observations from the Global Precipitation Climatology Center (GPCC; Schneider et al., 2014) are also used in comparison. Because of the similarity between the two datasets, only results from the APHRODITE are showed in the following text.

Simulated sea surface temperature (SST) is evaluated against the Operational Sea Surface Temperature and Sea Ice Analysis (OSTIA Donlon et al., 2012). As coupled simulations are configured to represent present day climate, OSTIA date from 1982 to 2012 is used.

## 2.2 Met Office Unified Model and experiments

The Met Office Unified Model (MetUM) Global Atmosphere 6.0 (GA6; Walters et al., 2017) and Global Coupled model 2.0 (GC2; Williams et al., 2015) are used. GA6 includes a relatively new dynamical core, which significantly increases mid-latitude variability and increases variability in the tropics. GC2 couples GA6 with with an ocean model (Necleus for European Modelling of the Ocean (NEMO); Madec, 2008) and a sea-ice model (the Community Ice CodE and the Los Alamos Sea Ice Model (CICE); Hunke and Lipscomb, 2004) via the coupler OASIS3 (Valcke, 2013) on 3 hourly frequency. GC2 showed an

improvement from previous configurations, particularly in terms of modes of variability, e.g., mid-latitude and tropical cyclone intensities, the Madden-Julian Oscillation and El Niño Southern Oscillation (Williams et al., 2015).

Six MetUM simulations are used, which can be grouped into three pairs. Each pair includes an atmospheric-only simulation (A) and an atmosphere-ocean coupled simulation (C), which have the same atmospheric horizontal resolution. Three different atmospheric horizontal resolutions are configured, $192 \times 145$ (N96), $432 \times 325$ (N216) and $1024 \times 769$ (N512s). Therefore,

the six simulations used here are denoted as AN96, CN96, AN216, CN216, AN512 and CN512. The equivalent side length of the atmospheric grid along the longitude at the equator is 200km, 90km and 40km, respectively. Atmosphere models have 85 hybrid height levels in the vertical covering 0-85km (Hewitt et al., 2011). Ocean model uses 75 vertical levels and the OCRA025 tri-polar grid which has $0.25°$ resolution at the equator (Hewitt et al., 2011; Madec and Imbard, 1996). Periods of simulation are listed in Table 1. Most simulations match the period of ERA-Interim (1982-2012) except N512 simulations

which have a shorter simulation period (1992-2012).

## 2.3 Water Accounting Model-2layers

WAM-2layers is a moisture tracking model developed by van der Ent et al. (2013, 2014). WAM-2layers is based on the atmospheric water conservation equation and combines information of precipitation, evaporation, atmospheric circulation and moisture to determine sources or sinks of moisture originated from a specified region. In this study, WAM-2layers is applied

to back-track moisture sources of precipitation over EA in both ERA-Interim reanalysis and MetUM simulations. Daily precipitation from either reanalysis or simulations is fed into WAM-2layers, which is integrated backward using circulation and humidity information on model/pressure levels. Domain and magnitude of moisture source will be calculated. A detailed description about WAM-2layers and its setup over EA are given in Guo et al. (2019). Due to EA crosses several climatic zones and has inhomogeneous hydrological features, this region is first divided into five subregions according precipitation minus evapo-

ration and topography (Figure 1). These regions are southeastern EA (region 1), Tibetan Plateau (region 2), central-eastern EA (region 3), northwestern EA (region 4) and northeastern EA (region 5). A similar division was used in Guo et al. (2018), where detail discussion about the division is given.

## 3 Differences to observation/reanalysis

### 3.1 Precipitation

Figure 2 shows annual mean precipitation in APHRODITE, MetUM AN96 and ERA-Interim and biases against APHRODITE. AN96 captures major features of precipitation over EA, i.e., the south-north precipitation gradient, the precipitation maxima over Sichuan Basin and Southeastern China (Figure 2b). However, compared to APHRODITE, AN96 overestimates precipitation over Tibetan Plateau, Sichuan Basin and Southeastern China; underestimates precipitation over the southern slope of the Himalayas (Figure 2c). There are also biases over southern Asia, i.e., the Indian Peninsula, Bangladesh and the Indochinese Peninsula. These similarities and biases are also common in other simulations (Supplement Figure S1). Comparing ERA-Interim to APHRODITE (Figure 2d and e), ERA-Interim overestimates precipitation over southwestern China and Tibetan Plateau. These biases will affect moisture tracking accuracy over these regions. However, using ERA-Interim precipitation for moisture source tracking remains as a better option because it matches with other ERA-Interim variables, i.e., moisture fluxes and evaporation.

Aforementioned precipitation biases are also reflected in the seasonal and regional mean precipitation over EA subregions (Figure 3). Both ERA-Interim reanalysis and MetUM simulations overestimate precipitation over southeastern EA (region 1) with MetUM simulations have larger biases. Precipitation biases over Tibetan Plateau (region 2) is high in both reanalysis and simulations. With increase in horizontal resolution, precipitation biases in MetUM simulations decrease, especially, from low-resolution (N96) to medium-resolution (N216). This is related to better representation of topography in simulations with higher resolutions. A detailed analysis about resolution-related moisture source change will be given in Section 4.1. Precipitation biases over regions 3 and 4 are smaller in coupled simulations than that in atmospheric-only simulations, especially, in June-July-August (JJA) and in medium-/high-resolutions simulations. This difference is due to the fact that the strength of the western North Pacific subtropical high (WNPSH) is weaker in coupled simulations. The weak WNPSH in the coupled simulations reduces moisture transport from low-latitude (i.e., region 1) to mid-latitude (regions 3 and 4), therefore, reduces the positive precipitation biases (Figure 4f, h). The weak WNPSH in coupled simulations has also been identified in previous studies (Rodríguez et al., 2017).

Both increasing horizontal resolution and introducing air-sea coupling in MetUM can improve precipitation simulation over EA, however, these improvements cannot sufficiently correct precipitation biases against observation. To further investigate these biases, moisture sources of EA precipitation are tracked and compared against those from reanalysis.

### 3.2 Moisture source

As aforementioned, due to the inhomogeneity of EA precipitation, EA is divided into five subregions (as in Figure 1). Moisture source for each subregion is investigated separately. Figure 4 shows the annual mean moisture source and vertically integrated moisture flux calculated from ERA-Interim, as well as differences between AN96 and ERA-interim. Compared to ERA-Interim, AN96 takes up less moisture from low-latitudes but more from mid-latitudes for all EA subregions. These differences in moisture source are largely associated with differences in moisture fluxes (Figure 4b, d, f, h and j). In AN96, the cross-

equatorial flow along the Somali Jet is too weak but the mid-latitude westerly is too strong. The moisture flux over region 1 is too zonal, which is coexisted with a weak WNPSH (the cyclonic moisture flux anomaly shown on Figure 4b). It is difficult to separate the causal relationship between the strong zonal monsoon flow and the weak WNPSH. However, these differences cause less moisture being transported to mid-latitude EA subregions from low-latitude landmass, which causes a negative
moisture source change from the southeastern EA to regions 3, 4 and 5 (Figure 4f, h and j). Over Tibetan Plateau, AN96 takes up less moisture over the whole moisture source domain, except the local source over the eastern Tibet. This explains the less seasonal mean precipitation over Tibetan Plateau in all simulations (Figure 3b).

The local moisture source is measured using the precipitation recycling ratio. The precipitation recycling ratio is defined as the proportion of precipitation in the target region that is contributed from the evaporation over the same region. Figure 5
shows the annual cycle of the precipitation recycling ratio calculated from ERA-Interim and MetUM. MetUM can produces similar annual cycles and magnitudes of the precipitation recycling ratio over regions 1, 3 and 4 (Table 2), but overestimates the recycling ratio over regions 2 (summer and autumn) and 5 (spring and autumn). However, from maps of moisture source (Figure 4), we learn that the precipitation recycling ratios in simulations is not closely matches with the distribution of it in ERA-Interim.

The remote moisture source is first compared using its shape. Due to the number of datasets used in this study, it would be lengthy to show maps of moisture source one by one. Therefore, instead of showing maps of moisture source, mass centres of moisture source from different datasets are calculated and compared collectively. Figure 6 shows mass centres in DJF and JJA. Mass centres are measured using moisture sources that account for 80% of precipitation in target regions, similar to those in Figure 4. Mass centres have also been measured using threshold at 50% and 65% of precipitation, results are consistent and
not sensitive to the choice of threshold. As shown in Figure 6, mass centres of moisture source in simulations show consistent seasonal variations as in reanalysis. However, there are systematic differences between simulations and reanalysis as well as among simulations themselves.

Over region 1 JJA, mass centres of MetUM are located approximately $5°$ to the north compared to ERA-Interim (triangles in Figure 6a). Similar to difference in the annual mean moisture source (Figure 4b), this is due to the weak cross-equatorial
moisture transport over the tropical Indian Ocean (Figure 7e). Over the same region in DJF, mass centres of MetUM are located $10°$ to the west compared to ERA-Interim (circles in Figure 6a), which is related to the stronger DJF westerly moisture flux. Similar shifts of both JJA and DJF mass centres are also seen over regions 2 and 3 (Figure 6b, c).

The northward shift of JJA mass centres over regions 4 and 5 is less, as these mid-latitude regions are less impacted by the EA summer monsoon, therefore, by the moisture flux bias over the tropical Indian Ocean. Over regions 4 and 5 DJF, on the
other hand, mass centres of MetUM are located to the east compared to ERA-Interim, especially in high-resolution simulations, i.e., CN216, AN512 and CN512, in which the eastward deviation is as large as $30°$ along the longitude (circles in Figure 6d, e). Comparing CN512's moisture source over region 5 DJF to that of ERA-interim (Figure 7b, d and f), CN512 picks up more moisture from Pacific Ocean (Seas of Japan and Okhotsk); while in ERA-Interim, more moisture is picked up from west, especially over the Mediterranean Sea, Red Sea and Persian Gulf. There is a low-/high-resolution division among simulations.
More details about this division will be discussed in Section 4.3.

The remote moisture source is further divided into four sections (tropical sea, tropical land, extra-tropical sea and extra-tropical land), together with local moisture source (measured by precipitation recycling ratio), contributions (both annal and seasonal means) from these sections are listed in Table 3 for both simulations and reanalysis. In annual mean, simulations reproduce the primary moisture sources for each EA subregion, i.e., the tropical sea for region 1 and the extra-tropical land for regions 2, 3, 4 and 5. However, the contribution from tropical sea is smaller in all simulations, which reflects the aforementioned negative moisture source difference over the tropical Indian Ocean. Instead, the contribution from extra-tropical land is greater in all simulations. In seasonal mean, however, discrepancies between simulations and reanalysis are greater, even the primary moisture source is different (as boldface values highlighted in Table 3). These seasonal discrepancies will be discussed in Section 4.3.

## 4 Differences in moisture source due to the model resolution and air-sea coupling

In the previous section, diagnoses of both precipitation and moisture source show that MetUM simulations are systematically different from ERA-Interim. By increasing horizontal resolution or including coupling, the gap between simulations and reanalysis cannot be bridged. On the other hand, however, diagnoses also show variations/improvements in precipitation and moisture source with changes in both resolution and coupling. Therefore, changes in moisture sources due to model resolution and coupling are discussed; links from changes in moisture source to precipitation are made in the section.

### 4.1 Change with resolution

As shown in Figure 3b, over Tibetan Plateau, precipitation bias in simulations is reduced compared to ERA-Interim. By comparing moisture source, this reduction is due to a weaker simulated remote moisture source/flux (Figure 4d). Precipitation bias is further reduced with increase in horizontal resolution (Figure 3b), which is consistent with previous studies showing that the higher the horizontal resolution is, the more remote moisture is blocked by Himalayas (Curio et al., 2015). On the other hand, the precipitation recycling ratio increases with horizontal resolution (Figure 5b), which indicates that, with reduced remote moisture contribution, the local moisture source becomes more important over Tibet. Figure 8a, b and c show that this intensified local source locates mainly over eastern Tibet. This is because that the eastern Tibet has greater precipitation (Figure 2), and that the remote moisture is transported onto Tibet via meridionally orientated valleys along its southeastern boundary. With reduced remote moisture flux caused by increased resolution along its boundary, the moisture source for this region shifts from remote to local source. This is also demonstrated as opposite trends in tracked local evaporation (increase) and low-level wind (decrease) shown in Figure 8d.

### 4.2 Change with coupling

To investigate the impact of air-sea coupling on moisture source, we focus on region 1, where ocean is the major contributor (according to Table 3). Differences in moisture source over region 1 JJA between coupled and atmospheric-only simulations are shown in Figure 9. As shown in Figure 9a, b and c, whatever the horizontal resolution is, coupled simulations show

consistent differences against atmospheric-only simulations, which include a reduced moisture contribution from the Indian Ocean (Arabian Sea and Bay of Bengal) but an increased moisture contribution from the Pacific Ocean (South and East China Seas). For the reduced moisture source over the Indian Ocean, it is linked to the cold SST bias, which has been reported in previous studies (Marathayil et al., 2013), and which is demonstrated in Figure 9d. As shown in Figure 9d, the averaged SST over the Arabian Sea shows a consistent negative anomaly in coupled simulations (filled bars on the lefthand side). On the other hand, over the Pacific Ocean, there is not a consistent SST bias associated with the increased moisture source. Instead, there is a consistent increase in low-level zonal wind in coupled simulations (dots to on righthand side of Figure 9d). As mentioned in Section 3, this wind bias is due to the EA summer monsoon flow in coupled simulations being too zonal. Coexisted with this wind bias, is the weak WNPSH, which explains the cyclonic circulation anomaly over the southeast coast of EA. This cyclonic anomaly converges extra evaporation caused by positive zonal wind bias (via the wind-evaporation feedback, hollow bars on righthand side of Figure 9d) and increases the local moisture contribution over region 1.

Compared to atmospheric-only simulations, coupled simulations pick up less moisture from the Indian Ocean along summer monsoon flow but more from adjacent oceans due to a circulation difference. As a result, precipitation does not show obvious difference between coupled and uncoupled simulations averaged over region 1 (Figure 3a).

## 4.3   Shift of major moisture source over mid-latidute regions

In Section 3.2, it has been mentioned that simulated mass centres of moisture source for regions 4 and 5 are separated into two groups according to resolution (Figure 6d, e). A similar division also exists among simulations when identifying the major moisture source for these regions (Table 3). In Table 3, moisture contribution from different remote sections (tropical sea, tropical land, extra-tropical sea and extra-tropical land) and local source are estimated for precipitation over all EA subregions on both annual and seasonal scales. On annual scale, the major moisture source for region 1 is tropical sea, but it is extratropical land for other subregions. This result is consistent among reanalysis and simulations. On seasonal scale, however, results are inconsistent, especially, for regions 4 and 5 in DJF. For regions 4 and 5 in DJF, in ERA-Interim, there is a shift of the major moisture source from extratropical land to extratropical sea (boldface values in Table 3). This shift is partly due to that the frozen land surface over the Eurasian continent in DJF reduces its evaporation, and partly due to that the stronger mid-latitude westerly brings in moisture from saturated surfaces west of target regions, such as the Mediterranean Sea, Black Sea and Caspian Sea (Guo et al., 2019). In simulations, this shift from land to sea is captured by simulations with higher horizontal resolutions, i.e., CN216, AN512 and CN512. However, maps of moisture source (Figure 10) show that this shift is simulations is caused by the wrong reason. For region 5 in DJF, simulations pick up more moisture from adjacent Pacific Ocean but less moisture from water bodies to the west. This difference is greater in coupled simulations and in simulations with higher horizontal resolution. In DJF, the land moisture source plays a minor role, due to its frozen soil and therefore small evaporation. The mid-latitude circulation in DJF is also reasonably simulated in all MetUM simulations (figure is not shown). Figure 11 shows that this difference in the moisture source is rooted in SST bias. The negative SST bias over the Mediterranean Sea indicates an underestimation of evaporation and moisture source (Figure 10a, d and g); the positive SST bias over Seas of Japan and Okhotsk, especially within higher resolutions simulations, indicates an overestimation of moisture source over these regions.

Note that, the positive SST bias over the East Asian coast is enlarged with the increasing horizontal resolution, especially over the Sea of Okhotsk and North Pacific. The SST bias in coupled simulations explain the shift of moisture source from extratropical land to extratropical sea in coupled simulations like CN216 and CN512. However, it can not explain the shift in AN512 wherein there is no SST bias involved. Note that, a similar but with smaller magnitude change in moisture source (increase over Seas of Japan and Okhotsk) is also found in atmospheric-only simulations with increasing horizontal resolution. Considering the fact that the precipitation in region 5 DJF is small (Figure 3e), the small increase in moisture source can eventually shift the major moisture source in AN512.

## 5  Conclusions

In this study, moisture sources of East Asian (EA) precipitation simulated in a set of MetUM configurations are traced using the Water Accounting Model-2layers (WAM-2layers) and compared to that of ERA-Interim reanalysis. The purpose of this study is to understand the precipitation bias in the MetUM and to link this bias to biases in evaporation and moisture transport over the moisture source region. Six MetUM simulations are used here, AN96, CN96, AN216, CN216, AN512 and CN512, which include an atmosphere-only simulation and an air-sea coupled simulation on three different horizontal resolutions.

MetUM simulations can reasonably capture EA precipitation features but also show systematic biases against observations regardless of of horizontal resolution or air-sea coupling. These biases include overestimates precipitation over southeastern EA and Tibetan Plateau.

To trace moisture source for EA precipitation, EA was first divided into five subregions, each of which has a relative homogenous hydrological feature. These subregions include southeastern EA, Tibetan Plateau, central-eastern EA, northeastern EA and northeastern EA. MetUM simulations show agreement with ERA-Interim in terms of capturing annual cycle of precipitation recycling ratio, seasonal shifts of moisture source. However, systematic differences between simulations and reanalysis remain. MetUM captures less moisture from tropical sea but more from extratropical land, which are linked to an underestimated moisture transport from tropical Indian Ocean and an overestimated moisture transport from mid-latitude Eurasian continent. These differences in moisture sources can be used to explain precipitation differences between simulations and reanalysis.

Although increasing horizontal resolution can not bridge the gap between simulated and observational precipitation, improvement in precipitation does show, especially over the Tibetan Plateau. This is ascribed to a reduced remote moisture source and to an enhanced local moisture source over the eastern Tibet.

Although including air-sea coupling does not necessarily improve precipitation over EA, differences in moisture source indicate model biases due to biases in surface flux and atmospheric circulation. Over southeastern EA in JJA, coupled simulations take up less moisture from the Arabian Sea due to a persistent SST cold bias, but take up more moisture from the South China Sea due to a positive wind-evaporation feedback and a cyclonic circulation anomaly. These differences in moisture source have similar magnitudes, which counteracts precipitation differences in coupled simulations when compared to atmospheric-only simulations.

Simulations with higher resolution and/or air-sea coupling, i.e., CN216, AN512 and CN512, capture a shift of the major moisture source over northwest and northeast EA in DJF. The major moisture source over these regions shifts from extratropical land to extratropical sea. However, the cause of this shift in simulations is different from that in reanalysis, and is mainly due to a positive anomaly of moisture source over the mid-latitude Pacific Ocean, which is related to the SST bias in the air-sea coupling and to the increase of the horizontal resolution.

ERA-Interim has been used here for its good performance on the EA precipitation (literature). However, considering that the source regions for EA precipitation are much larger, the accuracies of the hydrological components over other related regions are also important for correctly tracking moisture source. The hydrological variables (i.e., E-P) over the oceans show large discrepancies among the reanalysis products (Skliris et al., 2014), due to scarcity and discontinuity of observation over the oceans and due to (Schanze et al., 2010). In ERA-Interim, there is increasing E-P trend over the tropical Indian Ocean since 1979 compared to the observations indicating an increased net evaporation (Sliris et al., 2014). This bias could cause overestimation of moisture contribution from the tropical Indian Ocean provided that the circulation connecting this region and EA is less biased. Therefore, results shown here need to be interpreted with caution.

## 6 Discussions

In this study, we analysed systematic errors in EA precipitation simulated from a set of GCMs by linking these errors to errors in evaporation and moisture transports using a moisture tracking model. The advantage of using a moisture tracking model is that errors in evaporation, atmospheric moisture and circulation are combined and reflected in the tracked moisture source. Compared to previous studies that linked precipitation biases to the net moisture flux on the boundary of a study region, a moisture tracking model reveals more information on large spatial scale and from multiple hydrological components. Even though the precipitation bias could be small in some circumstance, method shown in this study can still reveal biases associated other hydrological components. As shown in current study, prior to precipitation, biases in surface flux and atmospheric circulation can cause moisture source biased toward opposite directions on different locations, even though the collective impact on precipitation is small due to the cancellation. These biases in surface flux and atmosphere circulation indicate that simulations have yet to improve their air-sea coupling and/or atmospheric forcings.

Moisture sources tracked using the WAM-2layers and the physical processes that link the source regions with the precipitation over EA have been discussed in Guo et al. (2019). Compared with studies employing other moisture methods, the results are consistent (Sun and Wang, 2015; Baker et al., 2015; Chu et al., 2017). As also shown herein, the Indian Ocean provides the largest portion of moisture during boreal summer for precipitation over southeast EA. This contribution to precipitation decreases with the latitude of precipitation. Meanwhile, the contribution from land increases increases with latitude. Local evaporation makes a larger contribution over the Tibetan Plateau compared to other EA subregions. During the boreal winter, due to the prevailing westerly and the frozen soil over the Eurasian continent, the Mediterranean Sea and other adjacent water bodies become the major moisture contributor for precipitation over the mid-latitude EA subregions. MetUM simulations can

generally capture most of these contributions, albeit biases are noticeable and vary with resolution and coupling. Similar biases have also been reported in Peatman and Klingaman (2018); Stephan et al. (2017a, b).

ERA-Interim is employed here for evaluating the simulations. It is chosen for its small residual in the global hydrological budget, its accurate representation of the mean and interannual variability of EA monsoon precipitation and its resemblance to the observation of evaporation over China (Trenberth et al., 2011; Lin et al., 2014; Sun and Wang, 2015). However, ERA-Interim has noticeable biases in the representation of the water cycle over the ocean, i.e., the P-E interannual variability in the tropical Indian Ocean is not well represented compared to observations (Skliris et al., 2014; Schanze et al., 2010). This bias could potentially affect the moisture contribution from the Indian Ocean estimated with ERA-Interim. To deliver more accurate information on the performance of MetUM in terms of tracking moisture sources, multiple reanalysis datasets should be included, so that biases from any single reanalysis dataset can be identified and considered.

In current climate modelling community, tools that can separately correct biases in air-sea coupling or atmospheric forcings are readily existed. For example, coupling a mixed-layer ocean model to an atmosphere model to correct surface flux (Hirons et al., 2015); adding a relaxation term to circulation variables to correct atmospheric circulation according to observations (Rodríguez et al., 2017). Moisture source associated biases, therefore, can serve as a guideline about where should correcting techniques be applied in simulations. Take the case of precipitation over southeastern EA in JJA as example, the surface flux correction should be applied over northern Indian Ocean to correct the cold SST bias, and the atmospheric circulation correction should be applied over western Pacific Ocean to correct the weak subtropical high bias. Although the deployment of these corrections are based on tracked moisture source over a small region, it could potentially correct simulations on a much large region as components of hydrological cycle are closely linked and coupled with the energy cycle via circulation and moisture transport. Therefore, we could expect improvement in precipitation over larger regions, i.e., Tibet and mid-latitude East Asia.

*Code and data availability.* For WAM-2layers, the model code is available at https://github.com/ruudvdent/WAM2layersPython/tree/distance/. For MetUM, the code is available only under license from the Met Office. The data used to produce the figures in this study have been published at https://doi.org/10.6084/m9.figshare.12801278.

*Author contributions.* LG and NPK designed the moisture tracking experiments. RJE developed WAM-2layers and LG adopted it to MetUM simulations. PLV and MED ran the MetUM similations. LG prepared the manuscript with contributions from all co-authors.

*Competing interests.* The authors declare that they have no conflict of interest.

*Acknowledgements.* This work and its contributors (LG, MED, PLV, AGT, AC) were supported by the UKChina Research and Innovation Partnership Fund through the Met Office Climate Science for Service Partnership (CSSP) China as part of the Newton Fund. LG was also funded by the UK National Centre for Atmospheric Science Visiting Scientist Programme. RJE acknowledges the Innovational Research Incentives Scheme with project number 016.Veni.181.015, which is financed by the Netherlands Organisation for Scientific Research (NWO). RJE also acknowledges funding from the Netherlands Organization for Scientific Research (NWO), project number 016.Veni.181.015. NPK was also funded by a UK Natural Environment Research Council Independent Research Fellowship (NE/L010976/1).

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

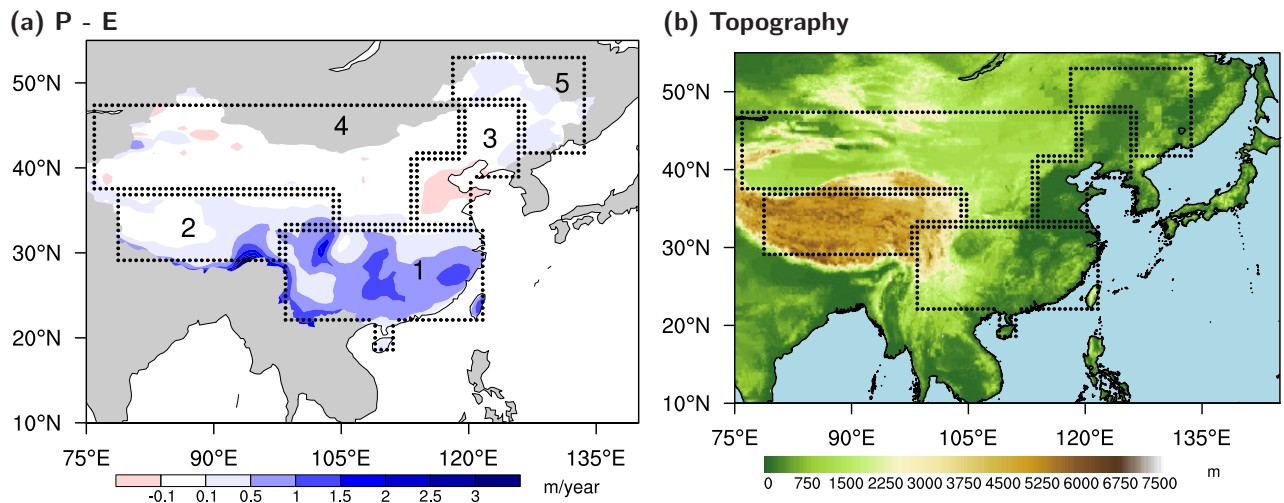

**Figure 1.** (a) Annual mean precipitation minus evaporation ($P - E$), calculated using ERA-Interim re-analysis during 1979–2016, units: m/year; (b) Topography over the EA landmass, units: m. Boxes 1–5 in (a) indicate subregions over EA. This is reproduced from Guo et al. (2018).

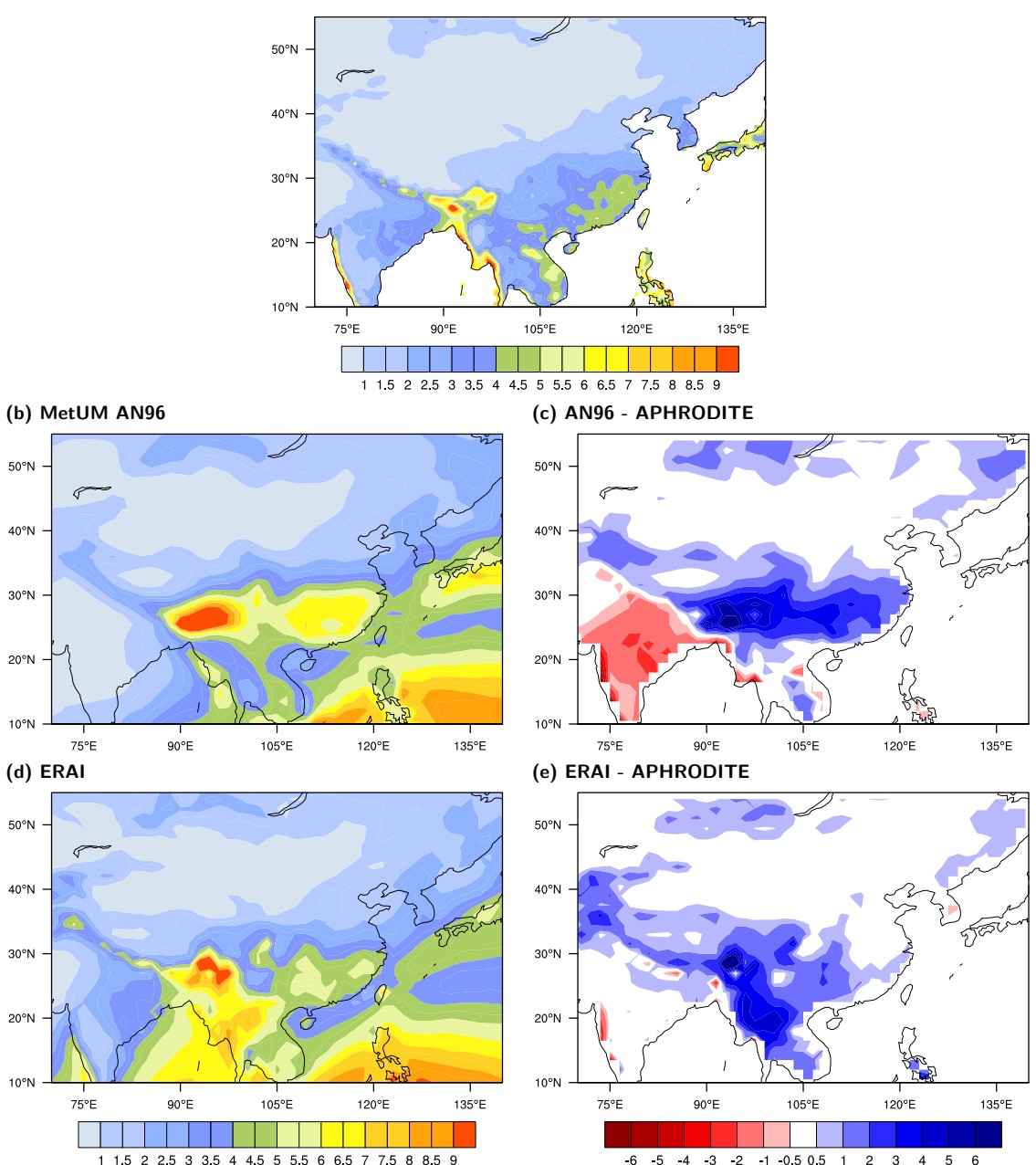

**Figure 2.** Annual mean precipitation of (a) APHRODITE, (b) MetUM AN96 and (d) ERA-Interim, and differences (c) between AN96 and APHRODITE, (e) between ERA-Interim and APHRODITE. The annual precipitations are averaged over 1982-2012. Units: mm · day$^{-1}$.

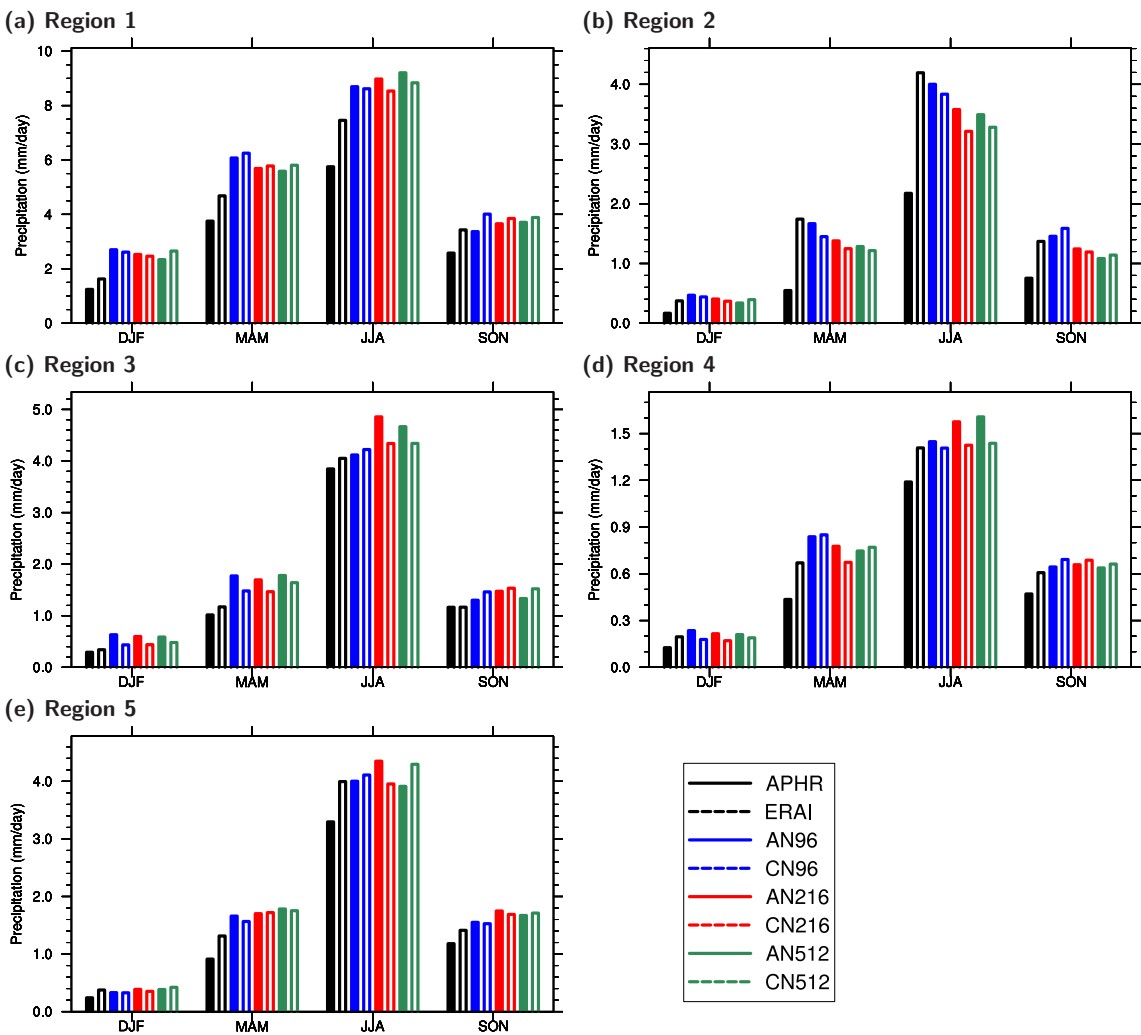

**Figure 3.** Seasonal and regional mean precipitation over EA subregions. Compared datasets include APHRODITE, ERA-Interim, AN96, CN96, AN216, CN216, AN512 and CN512. Units: $\mathrm{mm \cdot day^{-1}}$.

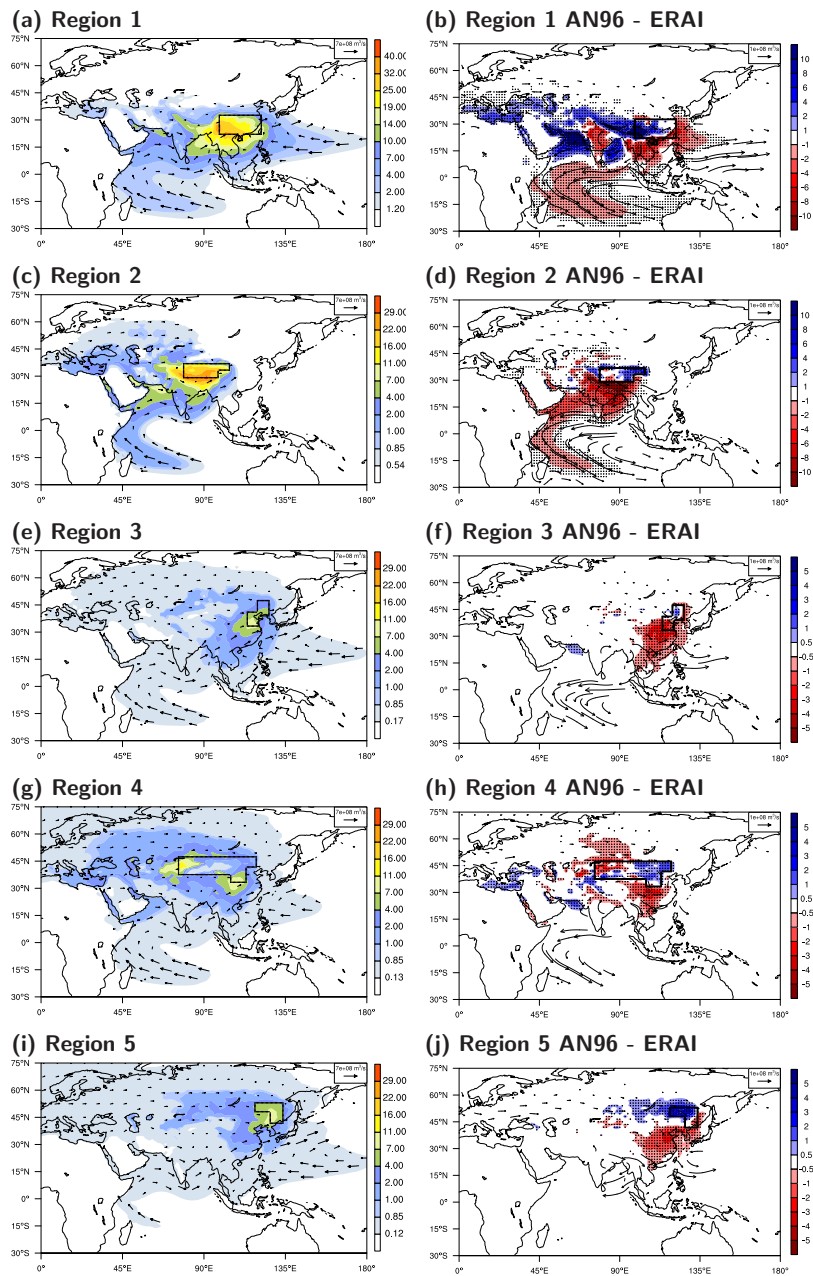

**Figure 4.** Annual mean moisture source for EA subregions (a, c, e, g and i, units: $mm \cdot month^{-1}$) and vertically integrated moisture flux (vector, units: $m^3 \cdot s^{-1}$) calculated from ERA-Interim. Moisture source accounts for 80% of precipitation is shown. Difference in annual mean moisture sources between AN96 and ERA-Interim (b, d, f, h and j). Units: $mm \cdot month^{-1}$. Black box in each panel indicates the target region.

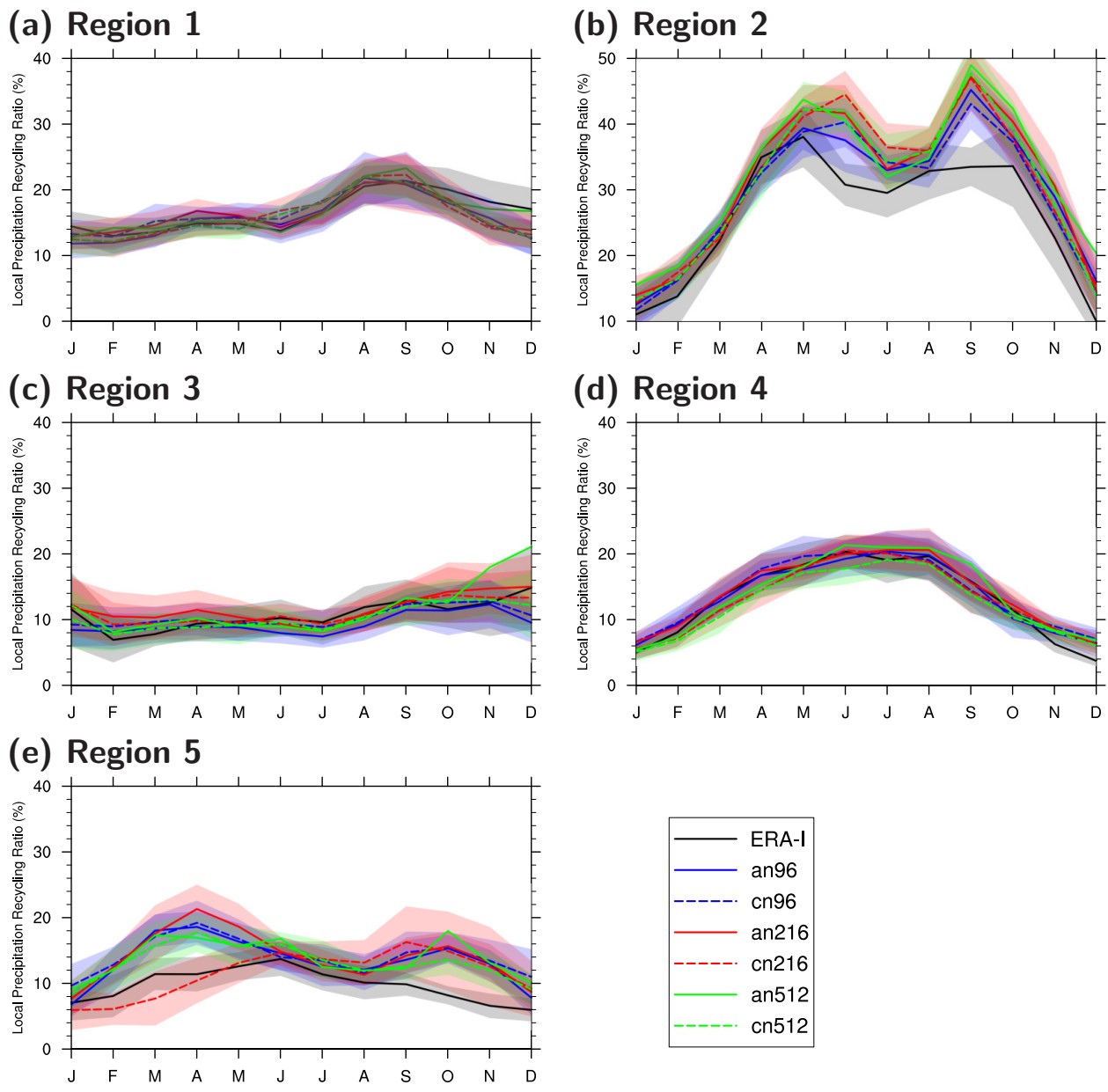

**Figure 5.** Annual cycle of the mean precipitation recycling ratio for EA subregions calculated from ERA-Interim and simulations, units: %. Shaded bands represent $\pm 1\sigma$.

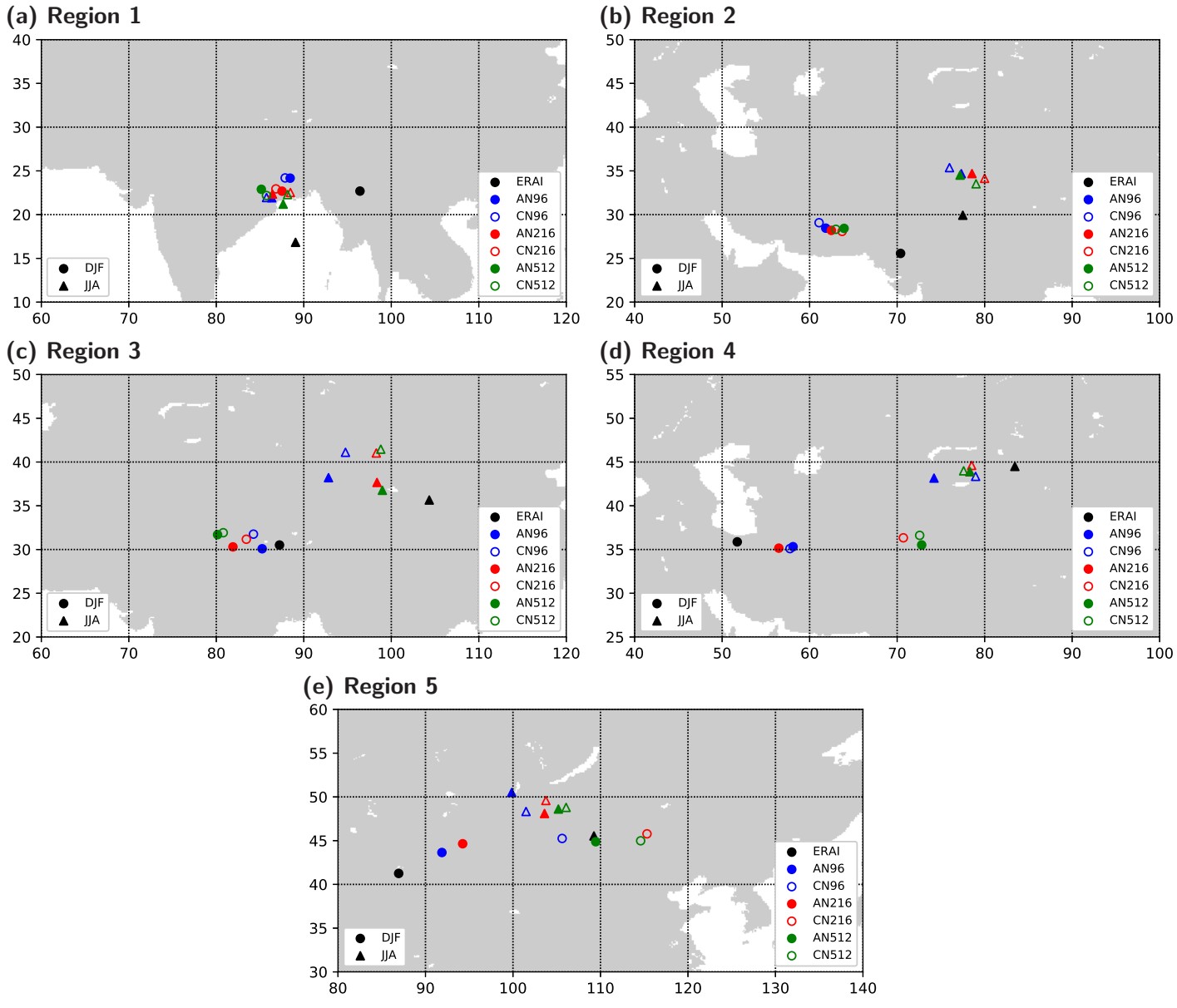

**Figure 6.** Mass centres of moisture source in DJF and JJA for regions 1-5 from ERA-Interim and MetUM simulations.

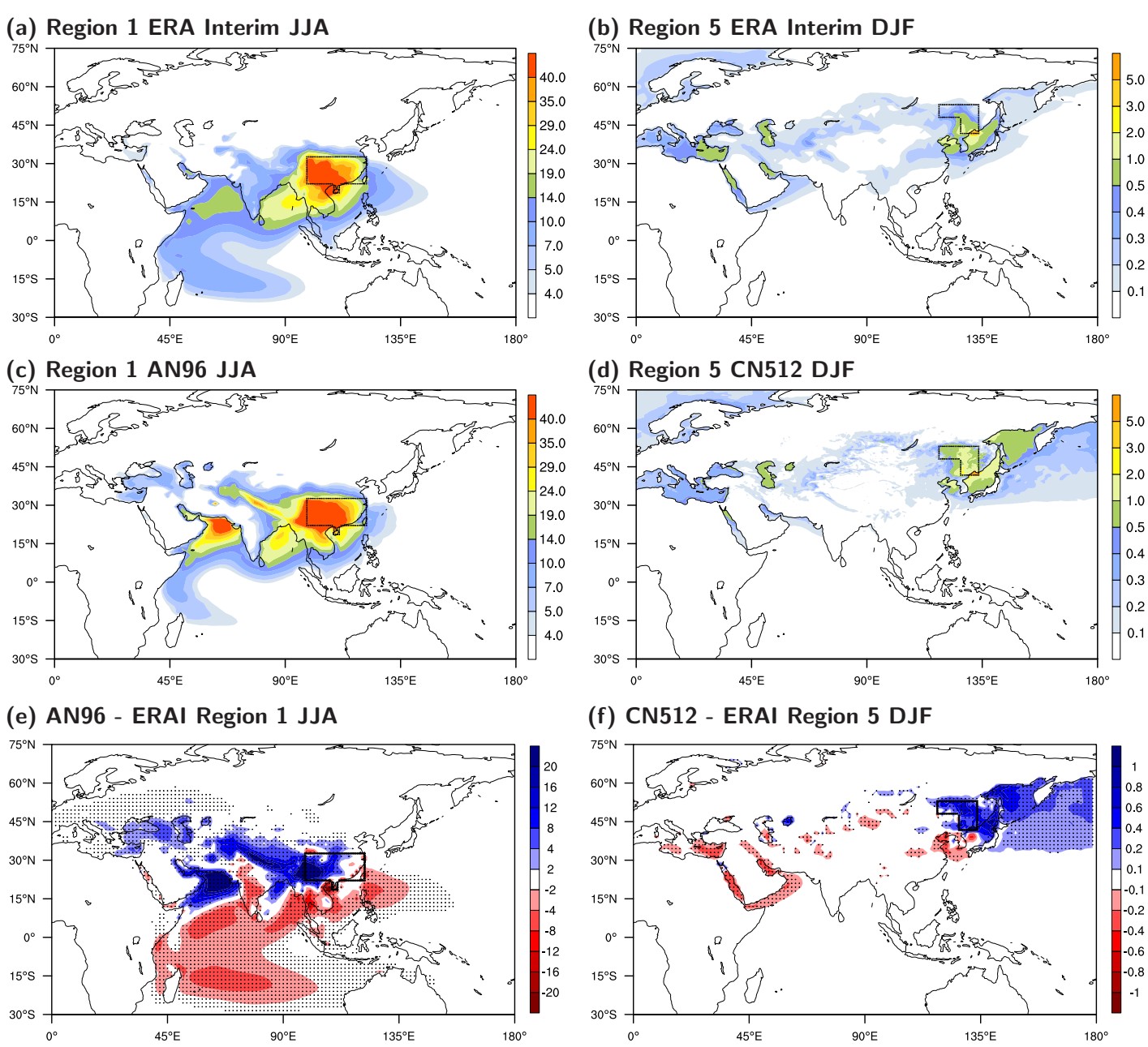

**Figure 7.** Moisture source during JJA for region 1 from ERA-Interim (a) and AN96 (c). Moisture source during DJF for region 5 from ERA-Interim (b) and CN512 (d). Difference of moisture source in region 1 JJA (e) between AN96 and ERA-Interim (f) between CN512 and ERA-Interim. Units: $mm \cdot month^{-1}$. The black box each panel represents target regions. Details of devision can be found in Figure 1.

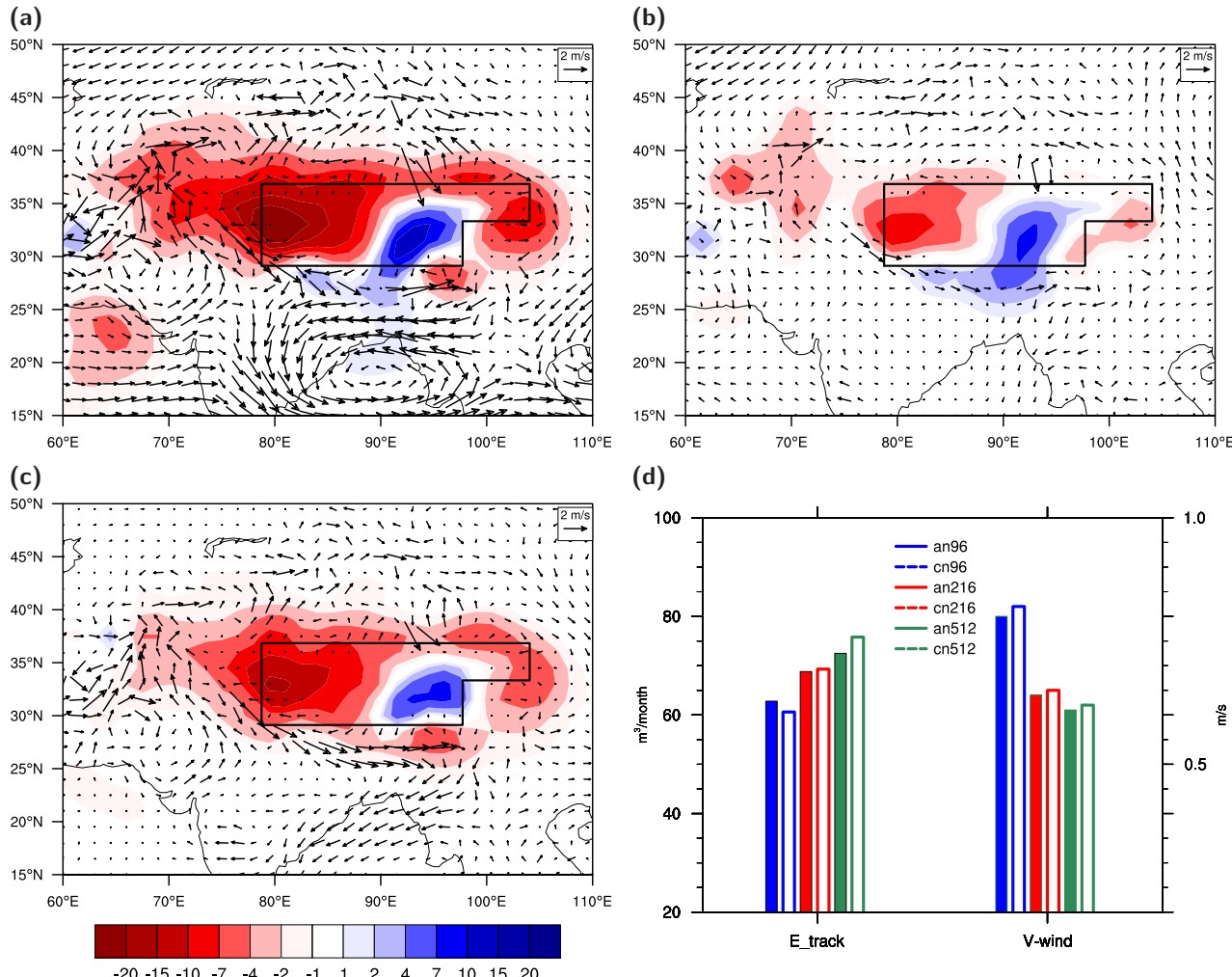

**Figure 8.** Difference of tracked evaporation (colour, units: $mm \cdot month^{-1}$) and 700 hPa wind (vector, units: $m \cdot s^{-1}$) in JJA over Tibetan Plateau (region 2) between (a) CN512-AN96, (b) CN512-CN216 and (c) CN216-CN96. (d) Seasonal mean tracked evaporation (*E_track*, $m^3 \cdot month^{-1}$) over eastern Tibet and the 700 hPa meridional wind (*V-wind*, $m \cdot s^{-1}$) along the southern boundary of the eastern Tibetan Plateau.

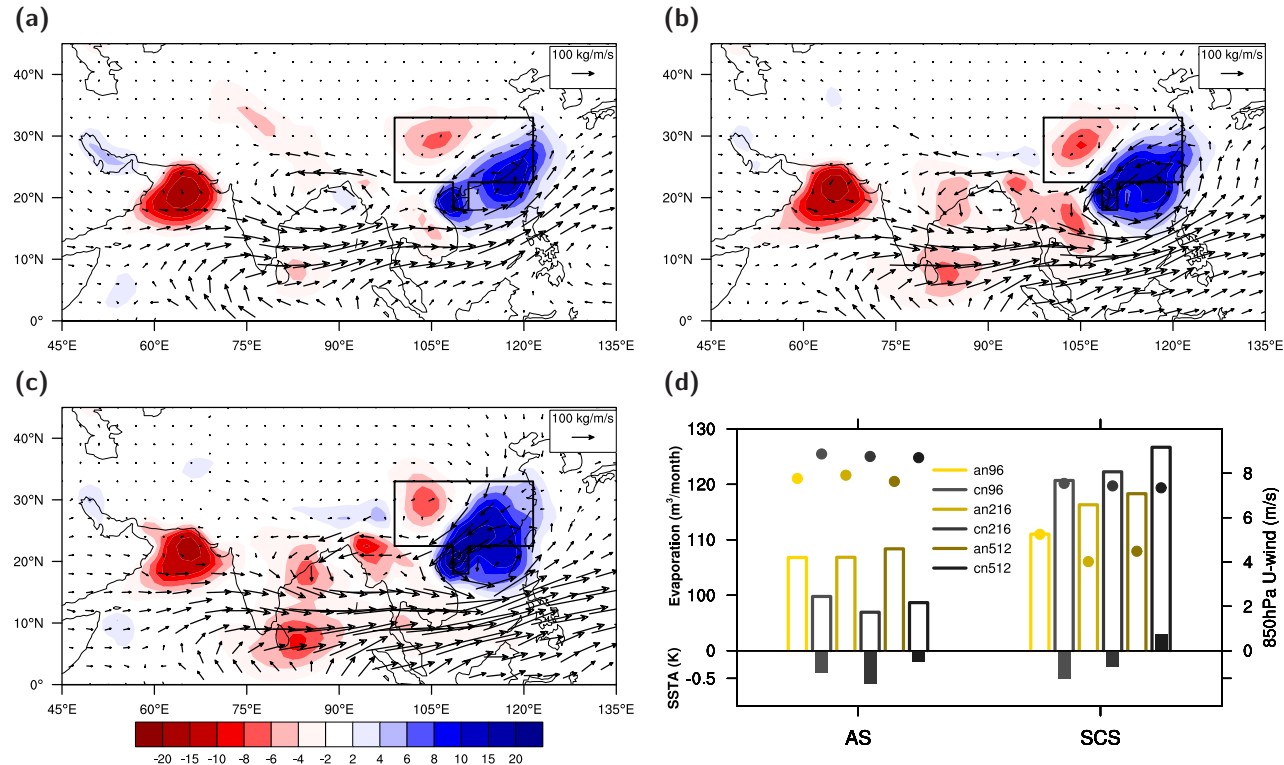

**Figure 9.** Differences in moisture source for precipitation over region 1 JJA between air-sea coupled and atmosphere-only simulations: (a) CN96 minus AN96, (b) CN216 minus AN216 and (c) CN512 minus AN512. Units: $mm \cdot month^{-1}$. Vectors are differences in the vertically integrated moisture flux, units: $kg \cdot m^{-1} \cdot s^{-1}$. (d) Mean evaporation (bars with outline-only, $m^3 \cdot month^{-1}$), mean zonal wind (dot, $m \cdot s^{-1}$) and sea surface temperature anomaly from observation (filled bar, units: K) over the Arabian Sea (AS) and South China Sea (SCS).

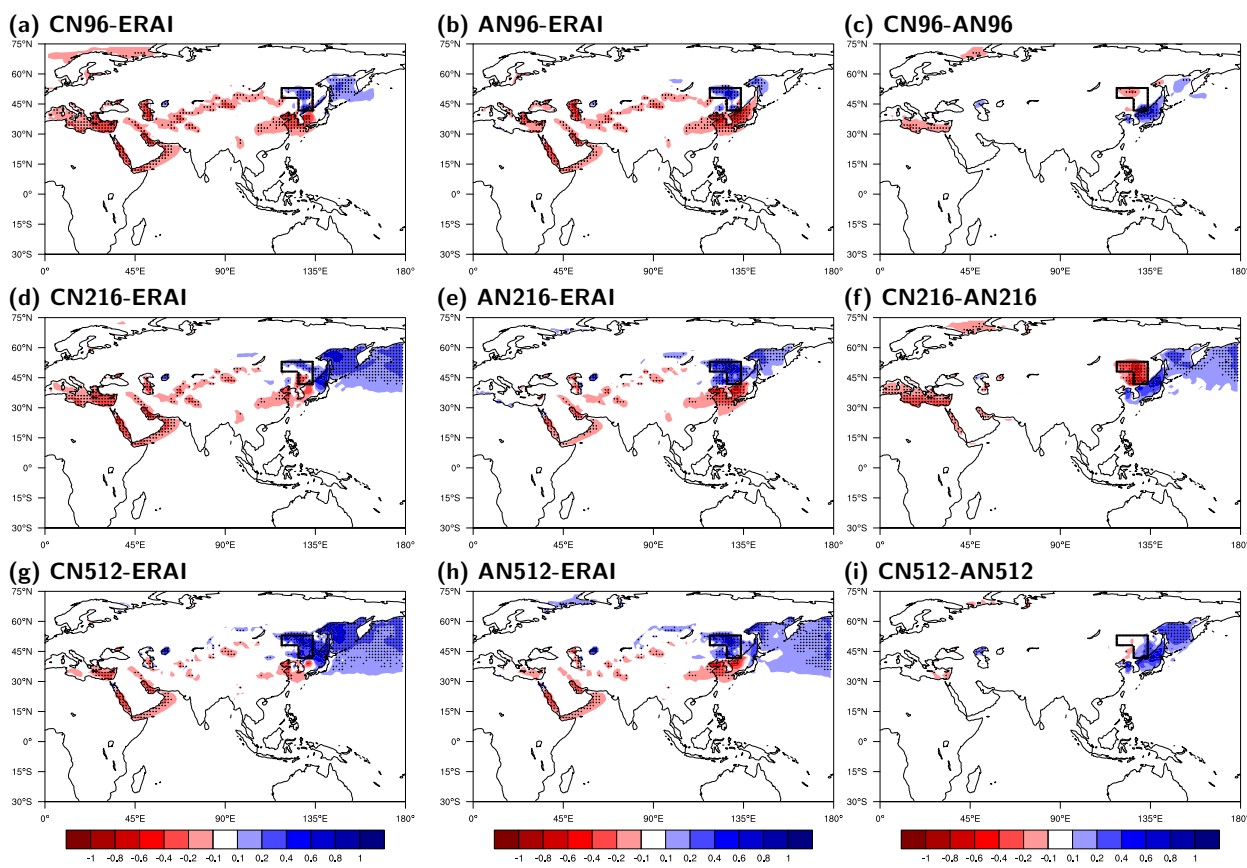

**Figure 10.** Difference in moisture source for region 5 in DJF. Units: mm/month.

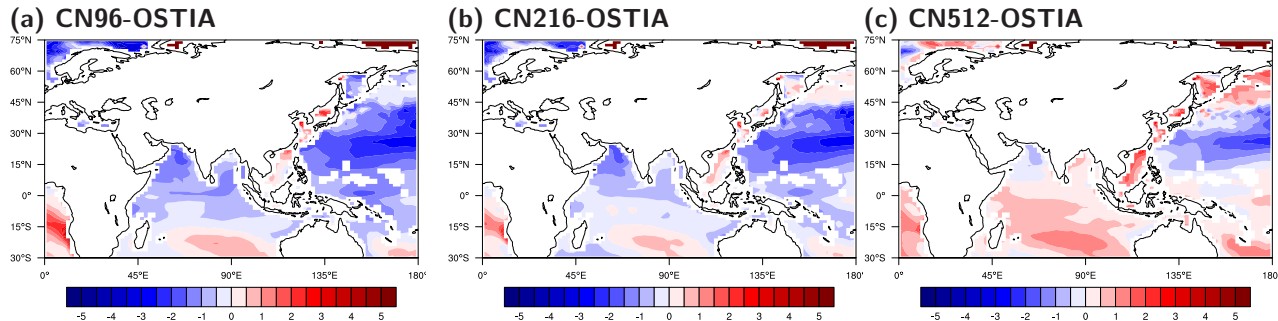

**Figure 11.** SST bias in MetUM coupled simulation in DJF. Units: K.

**Table 1.** Simulations used in this study. L85 of the atmospheric vertical resolution is a terrain-following hybrid height coordinate (units: m) that has 85 levels and a fixed model lid at 85km (Hewitt et al., 2011). L75 of the oceanic vertical resolution is a z*-coordinate (units: m) (Hewitt et al., 2011; Madec and Imbard, 1996).

| Simulation | Atmosphere | | Ocean | | Period |
|---|---|---|---|---|---|
| | horizontal | vertical | horizontal | vertical | |
| AN96 | $1.875° \times 1.25°$ | L85 | - | - | 1982-2012 |
| CN96 | $1.875° \times 1.25°$ | L85 | $0.25°$ | L75 | 31yr present day |
| AN216 | $0.56° \times 0.83°$ | L85 | - | - | 1982-2012 |
| CN216 | $0.56° \times 0.83°$ | L85 | $0.25°$ | L75 | 31yr present day |
| AN512 | $0.35° \times 0.23°$ | L85 | - | - | 1992-2012 |
| CN512 | $0.35° \times 0.23°$ | L85 | $0.25°$ | L75 | 21yr present day |

**Table 2.** Root-mean-square deviation of monthly precipitation recycling ratio (%) measured between MetUM simulations and ERA-Interim over five EA subregions.

|     | AN96 | CN96 | AN216 | CN216 | AN512 | CN512 |
|-----|------|------|-------|-------|-------|-------|
| cn1 | 1.8  | 1.8  | 1.7   | 2.0   | 2.0   | 1.9   |
| cn2 | 5.2  | 4.8  | 6.4   | 6.6   | 4.9   | 6.3   |
| cn3 | 1.7  | 1.6  | 1.8   | 1.3   | 1.5   | 1.3   |
| cn4 | 1.4  | 1.6  | 1.3   | 1.5   | 1.8   | 1.7   |
| cn5 | 4.6  | 5.0  | 5.3   | 3.3   | 2.8   | 4.0   |

**Table 3.** Annual and seasonal mean contributions to regional precipitation (regions 1-5, units: %) from the tropical ocean ($T_s$), tropical land ($T_l$), extratropical ocean ($X_s$), extratropical land ($X_l$) and local precipitation recycling ($\rho$). The solstices latitudes (23.4°N and 23.4°S) are used to separate the tropics and the extra-tropics. In each column, seven values represent seven datasets, shown in the order as ERA-Interim, AN96, CN96, AN216, CN216, AN512 and CN512. Values with boldface highlight difference that is discussed in Section 4.3.

| Region | Season | $T_s$ | $T_l$ | $X_s$ | $X_l$ | $\rho$ |
|---|---|---|---|---|---|---|
| 1 | Ann | 49 35 35 39 37 39 40 | 11 18 19 14 13 11 12 | 9 5 5 8 10 12 11 | 15 25 25 23 23 21 21 | 16 16 17 16 17 17 16 |
| | DJF | 40 37 36 41 39 42 39 | 15 21 22 15 15 12 11 | 15 8 6 12 15 17 19 | 17 23 23 20 19 17 18 | 14 12 13 13 12 12 12 |
| | MAM | 43 36 34 39 38 41 40 | 15 20 22 15 14 12 13 | 7 4 3 6 9 11 10 | 20 25 26 24 24 22 22 | 14 15 15 16 14 14 15 |
| | JJA | 55 36 36 39 37 39 41 | 10 17 17 14 12 11 13 | 6 4 4 6 7 8 8 | 12 26 26 24 25 24 22 | 17 17 18 17 19 19 16 |
| | SON | 47 31 33 37 35 36 36 | 8 16 18 13 13 11 12 | 13 9 9 11 13 15 14 | 12 26 22 21 21 19 20 | 20 19 18 18 18 18 19 |
| 2 | Ann | 25 9 9 9 10 10 10 | 6 5 6 4 4 3 3 | 7 4 4 8 10 13 12 | 30 48 47 44 40 38 40 | 31 34 34 36 37 36 35 |
| | DJF | 27 22 22 24 22 23 22 | 10 12 14 8 7 4 5 | 17 9 9 18 25 32 31 | 34 42 41 35 32 27 26 | 12 15 14 15 15 14 16 |
| | MAM | 17 11 9 11 13 13 13 | 6 7 8 5 5 4 4 | 8 4 4 7 12 16 15 | 34 44 45 40 35 32 32 | 34 34 34 36 35 35 36 |
| | JJA | 28 6 6 6 6 7 7 | 6 3 4 2 2 2 2 | 6 3 3 5 6 8 8 | 30 53 52 50 47 47 48 | 31 35 35 36 38 37 35 |
| | SON | 28 10 10 11 11 11 10 | 8 5 7 4 5 4 3 | 7 5 5 9 11 14 13 | 26 39 40 34 33 29 31 | 32 41 39 43 41 42 43 |
| 3 | Ann | 28 18 14 19 16 16 20 | 7 10 8 7 6 4 7 | 12 7 7 10 13 16 14 | 43 57 61 53 55 55 51 | 10 9 10 10 10 10 9 |
| | DJF | 19 21 18 20 18 17 15 | 10 14 15 9 9 5 5 | **30** 11 12 20 **28 36 34** | **31** 45 46 40 **34 33 37** | 10 8 9 11 11 9 9 |
| | MAM | 21 17 12 16 13 13 17 | 9 11 9 8 5 4 5 | 9 5 5 8 13 16 14 | 52 58 64 58 59 59 56 | 9 9 10 10 9 9 8 |
| | JJA | 33 20 16 22 19 18 24 | 6 9 7 8 5 4 7 | 9 6 7 8 9 11 9 | 42 57 61 53 57 58 51 | 10 8 9 9 10 9 8 |
| | SON | 22 13 10 14 13 13 15 | 5 7 8 6 6 4 5 | 20 12 10 16 20 24 22 | 41 57 60 51 49 47 47 | 12 11 12 13 13 12 11 |
| 4 | Ann | 15 9 8 8 8 8 9 | 4 6 6 4 3 2 3 | 11 6 6 10 14 17 16 | 54 63 64 61 58 57 55 | 17 17 17 17 16 15 16 |
| | DJF | 15 11 11 11 11 10 10 | 6 9 10 6 4 2 3 | **33** 14 13 25 **38 46 45** | **39** 58 58 51 **40 35 37** | 6 8 8 8 7 6 6 |
| | MAM | 11 9 6 8 10 9 11 | 5 7 7 5 3 3 3 | 10 5 5 9 16 19 18 | 58 63 64 62 56 54 54 | 17 16 18 17 16 15 14 |
| | JJA | 16 8 7 8 7 6 9 | 4 4 4 3 2 2 3 | 6 4 4 6 8 9 10 | 55 64 65 62 63 64 59 | 20 20 20 20 20 19 20 |
| | SON | 15 10 10 9 10 9 9 | 5 7 8 4 4 4 3 | 17 9 8 15 18 22 23 | 50 62 62 59 56 53 52 | 13 12 12 13 12 12 13 |
| 5 | Ann | 14 7 6 7 7 7 8 | 3 3 3 2 2 2 2 | 15 8 8 11 14 16 16 | 58 69 70 65 65 62 61 | 11 14 14 14 13 14 13 |
| | DJF | 8 5 4 5 4 4 4 | 3 4 4 3 2 1 1 | **49** 28 32 39 **54 56 54** | **34** 54 49 44 **34 28 31** | 7 9 11 10 6 10 9 |
| | MAM | 9 3 3 3 5 5 5 | 3 2 3 2 2 1 2 | 14 6 6 8 14 16 15 | 62 71 71 67 68 62 63 | 12 17 17 19 11 16 15 |
| | JJA | 18 8 7 10 8 9 10 | 3 3 3 3 2 2 3 | 9 6 6 7 7 9 9 | 59 70 72 67 69 67 66 | 11 13 13 13 13 14 12 |
| | SON | 10 5 5 5 5 5 8 | 2 2 2 2 2 1 2 | 25 13 11 16 20 25 26 | 54 66 67 62 58 55 53 | 9 14 14 14 14 13 12 |