# Peer review of "Effects of horizontal resolution and air-sea coupling on simulated moisture source for East Asian precipitation in MetUM GA6.0/GC2.0"

_Geoscientific Model Development, 2020_

## Short Comment (SC1) · 25 May 2020

Dear authors,

in my role as Executive editor of GMD, I would like to bring to your attention our Editorial version 1.2:

https://www.geosci-model-dev.net/12/2215/2019/

This highlights some requirements of papers published in GMD, which is also available on the GMD website in the 'Manuscript Types' section:

http://www.geoscientific-model-development.net/submission/manuscript_types.html

In particular, please note that for your paper, the following requirements have not been met in the Discussions paper:

- The main paper must give the model name and version number (or other unique identifier) in the title.

- "Code must be published on a persistent public archive with a unique identifier for the exact model version described in the paper or uploaded to the supplement, unless this is impossible for reasons beyond the control of authors. All papers must include a section, at the end of the paper, entitled "Code availability". Here, either instructions for obtaining the code, or the reasons why the code is not available should be clearly stated. It is preferred for the code to be uploaded as a supplement or to be made available at a data repository with an associated DOI (digital object identifier) for the exact model version described in the paper. Alternatively, for established models, there may be an existing means of accessing the code through a particular system. In this case, there must exist a means of permanently accessing the precise model version described in the paper. In some cases, authors may prefer to put models on their own website, or to act as a point of contact for obtaining the code. Given the impermanence of websites and email addresses, this is not encouraged, and authors should consider improving the availability with a more permanent arrangement. Making code available through personal websites or via email contact to the authors is not sufficient. After the paper is accepted the model archive should be updated to include a link to the GMD paper."

Therefore please name the MetUM including its version number (as named in Sect. 2.2) in the title of your article and provide your code publicly or provide the reasons why the code can not be made publicly available in your revised submission to GMD.

Yours,

Astrid Kerkweg

---

## Referee Comment (RC1) · Anonymous Referee #1 · 1 Jun 2020

General Comments: East Asian precipitation simulation is one of the great challenges faced by climate scientists due to the complexity of East Asian climate system and topography. The simulation of East Asian precipitation is sensitive to model resolution and air-sea coupling. This paper investigated the moisture sources of East Asian precipitation simulated by MetUM models using WAM-2layers. It provides a novel way to understand model bias. This study shows evidences about the sensitivities of moisture sources of EA precipitation to model horizontal resolution and air-sea coupling. The results are convincing, and helpful for model developers and climate model users. This paper is well structured and written. Thus, I suggest a minor revision. Specific Comments: 1. As shown in Fig3d, the moisture source over the tropics in region1

and region2 is underestimated, and more source from mid-latitude is transported to the two regions. Is there any coupling between the biases of the tropical source and mid-latitude source? 2. It would be useful to examine the travelling time and distance of moisture to further check the model biases and sensitivities to resolution and air-sea coupling. 3. As for the moisture bias of region5 precipitation in DJF, it shows that less moisture source from the mid-latitude and more moisture source from the Seas of Japan and Okhotsk lead to the eastward shift of the moisture center. This paper well discussed the positive anomalies of the moisture source from east of region5 with resolution. How about the contribution of mid-latitude circulation or evaporation bias?

Typing errors 1. Fig.10i, "CN512-CN512"-> "CN512-AN512" 2. P8 L229 Seas of Japan ad Okhotsk-> Seas of Japan and Okhotsk P10 L289 cecessary->necessary

---

## Referee Comment (RC2) · Anonymous Referee #2 · 2 Jun 2020

The paper deals with the sensitivity of climate models to grid resolution and atmosphere/ocean coupling in simulating moisture transported from ocean/land moisture sources ending up as precipitation over East Asia. The study is innovative and the subject itself is of great interest, especially for the climate modelling community. The manuscript is well introduced, well organised and well written. Their analysis of biases is well founded and findings are robust. However I have some concerns about the discussion section and the use of reanalysis and observational data which need to be addressed prior to publication (see major comments below).

Major points

- It is not clear what is the exact period used to calculate the climatological annual mean precipitation for MetUM simulations, ERA-interim and Aphrodite datasets. Is this the common matching period 1982-2007 (if MetUM AN & CN512 is included the common period would be limited to 1992-2007), or different periods i.e. 1979-2007 for Aphrodite, 1982-2012 for MetUM AN/CN 96/216, etc. If significant trends are present in these timeseries (which is the case over several regions of East Asia in the Aphrodite timeseries) the choice of period may have significant impacts on the calculated climatological annual mean patterns. Ideally a common matching period for all datasets should be used, or, at least the associated inconsistencies when comparing annual mean precipitation patterns of products of different periods should be discussed in the text (i.e. in addition to inconsistencies related to AN/CN 512 shorter period simulations already discussed in the text).

- Along with Aphrodite, I could use an additional observationally-based dataset for land EA precipitation such as the CPC Unified Gauge-Based Analysis of Global Daily Precipitation (at 0.5 deg. resolution, available from 1979-present) which, in contrast with Aphrodite, is fully matching ERA-I and MetUM simulation periods, to estimate precipitation biases w.r.t. ERA-I and MetUM. Although these datasets are based more or less on the same gauge data stations, different interpolation methods to fill the gaps and different periods can have significant impacts on calculating the climatological mean pattern of precipitation.

- Discussion section 5.1 is too short. Although the paper is focusing on the impact of model grid resolution and air-sea coupling on biases in moisture transport from ocean/land moisture sources ending up as precipitation over East Asia more text could be included in the discussion section about representation of physical processes involved in moisture transport in East Asia in the reanalysis and MetUM simulations. You could briefly compare your findings with previous moisture source/transport diagnostic studies in East/Southeast Asia using Langragian models and reanalysis data (e.g. Sun and Wang, 2015; Baker et al. 2015; Chu et al. 2017).

- Although ERA-Interim shows indeed a good skill in simulating mean and inter-annual variations in land precipitation over East Asia this is not always the case for water cycling over the ocean. For example, P-E interannual variability in the tropical Indian Ocean is not well represented in ERA-Interim as compared to observationally-based products (see Skliris et al. 2014). This may affect the simulation of moisture transport from Indian Ocean moisture sources for SE Asia precipitation in ERA-Interim. In general there are large discrepancies between the different reanalyses in representing E & P variations over the ocean (see Schanze et al. 2010). A more critical discussion is needed in the text concerning the use of a single reanalysis product as a benchmark to compare moisture sources traced from climate model simulations.

- I would suggest to additionally use the ERA5 dataset (which replaced ERA-Interim a year ago) in your analysis which has much higher horizontal resolution (∼30km) and with considerable improvements w.r.t. ERA-interim including better global balance of precipitation and evaporation and better precipitation over land, especially in the tropical regions. In addition, this way you may also investigate the impact of higher model resolution on the reanalysis biases and compare changes due to increasing resolution in products with similar resolution in ECMWF and MetUM products (i.e. ERA-Interim/AN216 vs. ERA5/AN512). Although I recognise that this requires a considerable extra effort and while the paper is publishable in its current form, I think it could strengthen your analysis and further improve the robustness of your findings.

Minor points - You should provide the ERA-Interim space grid resolution in section 2.1

- I would suggest to use MetUM AN216 (rather than AN96) to compare with ERA-Interim in figures 2 & 4 as these two datasets have similar horizontal grid resolution

- Table 1: Please indicate units for horizontal grid resolution (degrees) and vertical resolution (levels)

Typing errors Line 274: " . . . the eastern Tibetan Plateau, where the sruface is wetter . . ." Change into "surface"

References Baker, A. J., H. Sodemann, J. U. L. Baldini, S. F. M. Breitenbach, K. R. Johnson, J. van Hunen, and P. Zhang (2015), Seasonality of westerly moisture transport in the East Asian summer monsoon and its implications for interpreting precipitation $\delta$18O. J. Geophys. Res. Atmos., 120, 5850–5862, doi:10.1002/2014JD022919.

Chu Q., Wang Q., Feng G. (2017), Determination of the major moisture sources of cumulative effect of torrential rain events during the preflood season over South China using a Lagrangian particle model, J. Geophys. Res. Atmos., 122, 8369–8382, doi:10.1002/ 2016JD026426.

Schanze JJ, Schmitt RW, Yu LL (2010) The global oceanic freshwater cycle: a state-of-the-art quantification. J Mar Res 68:569–595. doi:10.1357/002224010794657164

Skliris N., et al (2014) Salinity changes in the World Ocean since 1950 in relation to changing surface fresh water fluxes. Clim Dyn 43:709–736.

Sun B., and Wang H. (2015). Analysis of the major atmospheric moisture sources affecting three sub-regions of East China. Int. J. Climatol. 35: 2243–2257.

———————————————

---

## Author Comment (AC1) · 22 Aug 2020

Astrid Kerkweg

a.kerkweg@fz-juelich.de

Thank you for reminding us to comply with the GMD requirements. We will make the following modifications to the manuscript.

Dear authors,

in my role as Executive editor of GMD, I would like to bring to your attention our Editorial version 1.2: https://www.geosci-model-dev.net/12/2215/2019/

This highlights some requirements of papers published in GMD, which is also available on the GMD website in the 'Manuscript Types' section:

http://www.geoscientific-model-development.net/submission/manuscript_types.html

In particular, please note that for your paper, the following requirements have not been met in the Discussions paper:

• The main paper must give the model name and version number (or other unique identifier) in the title.

We have added the model name and version numbers to the title and abstract. The new title will be: "*Effects of horizontal resolution and air-sea coupling on simulated moisture source for East Asian precipitation in MetUM GA6.0/GC2.0*".

• "Code must be published on a persistent public archive with a unique identifier for the exact model version described in the paper or uploaded to the supplement, unless this is impossible for reasons beyond the control of authors. All papers must include a section, at the end of the paper, entitled "Code availability". Here, either instructions for obtaining the code, or the reasons why the code is not available should be clearly stated. It is preferred for the code to be uploaded as a supplement or to be made available at a data repository with an associated DOI (digital object identifier) for the exact model version described in the paper. Alternatively, for established models, there may be an existing means of accessing the code through a particular system. In this case, there must exist a means of permanently accessing the precise model version described in the paper. In some cases, authors may prefer to put models on their own website, or to act as a point of contact for obtaining the code. Given the impermanence of websites and email addresses, this is not encouraged, and authors should consider improving the availability with a more

permanent arrangement. Making code available through personal websites or via email contact to the authors is not sufficient. After the paper is accepted the model archive should be updated to include a link to the GMD paper."

We have made the code and data as available as possible. We will add the following information in the Code and Data availability Section: "*For WAM-2layers, the model code is available at https://github.com/ruudvdent/WAM2layersPython/tree/distance/. For MetUM, the code is available only under license from the Met Office. The data used to produce the figures in this study have been published at https://doi.org/10.6084/m9.figshare.12801278.*"

Therefore, please name the MetUM including its version number (as named in Sect.2.2) in the title of your article and provide your code publicly or provide the reasons why the code cannot be made publicly available in your revised submission to GMD.

Yours,

Astrid Kerkweg

---

## Author Comment (AC2) · 22 Aug 2020

General Comments: East Asian precipitation simulation is one of the great challenges faced by climate scientists due to the complexity of East Asian climate system and topography. The simulation of East Asian precipitation is sensitive to model resolution and air-sea coupling. This paper investigated the moisture sources of East Asian precipitation simulated by MetUM models using WAM-2layers. It provides a novel way to understand model bias. This study shows evidence about the sensitivities of moisture sources of EA precipitation to model horizontal resolution and air-sea coupling. The results are convincing, and helpful for model developers and climate model users. This paper is well structured and written. Thus, I suggest a minor revision.

We thank the reviewer for providing useful comments and discussions to help us improve this manuscript.

**Specific Comments:**

1. As shown in Fig3d, the moisture source over the tropics in region1 and region2 is underestimated, and more source from mid-latitude is transported to the two regions. Is there any coupling between the biases of the tropical source and mid-latitude source?

We believe that the reviewer meant Figure 4b&d instead of Figure 3d.

If the moisture flux from the tropics is weak as shown in MetUM, then the moisture for precipitation over regions 1 and 2 should come from elsewhere. In the case of region 1, the additional moisture comes from the mid-latitude regions; in the case of region 2, the additional moisture comes from local evaporation regions; the additional moisture comes from local evaporation.

With that being said, there is a positive precipitation anomaly over the tropical Indian Ocean in MetUM simulations (Figure R1a, enclosed with this response). Besides, the subtropical jets at 200hPa in both hemispheres shift southward in MetUM (Figure R1c). The anomalous monsoon westerly in the MetUM between 15°-30°N at 850hPa transports more moisture from the west, which is collocated with the positive moisture source anomaly shown in Figure 4b (Figure R1b). The anomalous circulation is consistent with the anomalous convection; however, it is difficult to separate the cause and effect without carrying out extra experiments.

For the Tibetan Plateau (region 2, Figure 4d), there is a positive mid-latitude moisture source anomaly within the region and to the east, which is collocated

with a positive evaporation anomaly in MetUM simulations. Here, we focus our analysis on the summer, as the differences shown in Figure 4 show the patterns of summer monsoon over Asia.

[Figure]

Figure R1. (a) Difference of JJA precipitation between MetUM AN96 and ERA-Interim averaged over 1982–2012 (mm/month). (b) Difference of JJA 850hPa wind between MetUM AN96 and ERA-Interim averaged over 1982–2012 (m/s). (c) Latitude-Pressure plot of the JJA U-wind climatology averaged between 60° and 90°E in ERA-Interim during 1982–2012 (contour, m/s) and the difference between MetUM AN96 and ERA-Interim over the same domain and period (colour, m/s).

2. It would be useful to examine the travelling time and distance of moisture to further check the model biases and sensitivities to resolution and air-sea coupling.

We thank the reviewer for the suggestion. We investigated both the travel time and travel distance of the tracked moisture. However, the results do not show a systematic change with either resolution or coupling. We are investigating the reason: one possible cause could be the delayed monthly mean, since all results shown in this manuscript are obtained from monthly mean outputs. The monthly mean is not simply calculated from the 1$^{st}$ of each month to the last day of the same month, because the moisture transport takes place over timescales of days and weeks. For example, moisture evaporated from the Mediterranean Sea typically takes 15 days to be transported to EA. Therefore, in

the backward tracking, any precipitation put back into the WAM-2layers over EA would take 15 days to reach the Mediterranean Sea. Therefore, the monthly mean moisture source for EA precipitation over the Mediterranean Sea starts from the 15[th] of the month and ends a month later (the exact date depends on the length of the month). We have treated the tracked moisture shown in this manuscript with this method, but we have not treated the travelling time and distance in the same way. We suspect that this causes the inconsistent results between tracked moisture and the travelling time or distance. To confirm this idea, we are re-running all our WAM tracking simulations, but as this process requires a substantial amount of computational time, we will need to report the results in a future study.

3. As for the moisture bias of region5 precipitation in DJF, it shows that less moisture source from the mid-latitude and more moisture source from the Seas of Japan and Okhotsk lead to the eastward shift of the moisture centre. This paper well discussed the positive anomalies of the moisture source from east of region5 with resolution. How about the contribution of mid-latitude circulation or evaporation bias?

As shown in Figure 10, over region 5 in DJF, the major moisture source biases across all MetUM simulations come from the mid-latitude water surfaces, i.e., the Seas of Japan and Okhotsk in the Pacific, the Mediterranean Sea, Caspian Sea, Red Sea and Persian Gulf in western and central Asia, not from the Eurasian land surface. This indicates that the MetUM bias of evaporation over the Eurasian continent is small (due to the frozen soil). The biases in the mid-latitude lower-tropospheric circulation during DJF are also small, as indicated by Figure R2 (below).

Similar to the moisture source bias caused by SST bias over the Seas of Japan and Okhotsk, the reduced moisture sources over the Mediterranean Sea, Caspian Sea, Red Sea and Persian Gulf are linked to the negative SST biases over those water bodies, especially in the low and mid-resolution coupled simulations.

The following revision will be made on Page 8 Lines 30-31: "*In DJF, the land moisture source plays a minor role, due to its frozen soil and therefore small evaporation. The mid-latitude circulation in DJF is also reasonably simulated in all MetUM simulations (figure not shown).*".

[Figure]

Figure R2. Difference in DJF moisture flux between MetUM simulations and ERA-Interim. Units: m³/month.

**Typing errors**

1. Fig.10i, "CN512-CN512"-> "CN512-AN512"

Correction has been done.

2. P8 L229 Seas of Japan ad Okhotsk-> Seas of Japan and Okhotsk. P10 L289 cecessary->necessary.

Corrections have been done.

---

## Author Comment (AC3) · 22 Aug 2020

The paper deals with the sensitivity of climate models to grid resolution and atmosphere/ocean coupling in simulating moisture transported from ocean/land moisture sources ending up as precipitation over East Asia. The study is innovative and the subject itself is of great interest, especially for the climate modelling community. The manuscript is well introduced, well organised and well written. Their analysis of biases is well founded, and findings are robust. However, I have some concerns about the discussion section and the use of reanalysis and observational data which need to be addressed prior to publication (see major comments below).

We thank the reviewer for the evaluation and comments. We have replied following each specific comment.

Major points

– It is not clear what is the exact period used to calculate the climatological annual mean precipitation for MetUM simulations, ERA-interim and Aphrodite datasets. Is this the common matching period 1982–2007 (if MetUM AN & CN512 is included the common period would be limited to 1992–2007), or different periods i.e. 1979–2007 for Aphrodite, 1982–2012 for MetUM AN/CN 96/216, etc. If significant trends are present in these timeseries (which is the case over several regions of East Asia in the Aphrodite timeseries) the choice of period may have significant impacts on the calculated climatological annual mean patterns. Ideally a common matching period for all datasets should be used, or, at least the associated inconsistencies when comparing annual mean precipitation patterns of products of different periods should be discussed in the text (i.e. in addition to inconsistencies related to AN/CN 512 shorter period simulations already discussed in the text).

As suggested by the reviewer, a common period of 1982–2012 is now used to calculate the precipitation climatologies for ERA-Interim, APHRODITE and MetUM simulations at N96 and N216 resolutions.

To accomplish this, the APHRODITE dataset has been extended from 2007 to 2012 using its product V1101EX-R1 obtained from the second phase of the APHRODITE project. According to its guidance (https://climatedataguide.ucar.edu/climate-data/aphrodite-asian-precipitation-highly-resolved-observational-data-integration-towards), this extension uses an algorithm consistent with that of the original dataset, but with added data and improved quality control.

We leave the precipitation climatology for the MetUM at N512 resolution for the period 1992–2012, as this is the period for which we have data and can track moisture for precipitation using WAM. Restricting all datasets to this period would greatly reduce the sample size for analysis in the other datasets (from 30 years to 20 years). Figure 2 in the revision has now been updated along with its caption. The climatological annual mean precipitation pattern in APHRODITE has not dramatically changed, as the pattern correlation coefficient between the old (1982–2007) and updated (1982–2012) periods is 0.99. In addition to Figure 2, data information in Section 2 is also updated accordingly.

Page 3 Lines 30–31 of the revised paper will read: "*To match with MetUM simulations, the period between 1982–2012 is used for both ERA-Interim and APHRODITE.*"

Page 4 Lines 18–20: "*Periods of simulation are listed in Table 1. Most simulations match the period of ERA-Interim (1982–2012) except N512 simulations which have a shorter simulation period (1992–2012).*"

– Along with Aphrodite, I could use an additional observationally-based dataset for land EA precipitation such as the CPC Unified Gauge-Based Analysis of Global Daily Precipitation (at 0.5 deg. resolution, available from 1979–present) which, in contrast with Aphrodite, is fully matching ERA-I and MetUM simulation periods, to estimate precipitation biases w.r.t. ERA-I and MetUM. Although these datasets are based more or less on the same gauge data stations, different interpolation methods to fill the gaps and different periods can have significant impacts on calculating the climatological mean pattern of precipitation.

Although we have matched the data availability of APHRODITE with ERA-Interim and the MetUM (see response to comment above), we have also followed the reviewer's suggestion to compare these products with an additional dataset. We chose another gauge-based gridded precipitation product, from the Global Precipitation Climatology Center (GPCC), which covers the same period and has a similar resolution to the CPC dataset. In terms of annual mean precipitation climatology, GPCC and APHRODITE are similar, with a pattern correlation coefficient of 0.89. With this information in mind, we continue to use APHRODITE in the rest of our manuscript.

The following modification will be added in the revisited manuscript on Page 3, Lines 31–33: "*Other precipitation observations from the Global Precipitation Climatology Center (GPCC; Schneider et al., 2014) are also used in comparison. Because of the similarity between the two datasets, only results from the APHRODITE are showed in the following text.*"

– Discussion section 5.1 is too short. Although the paper is focusing on the impact of model grid resolution and air–sea coupling on biases in moisture transport from ocean/land moisture sources ending up as precipitation over East Asia, more text could be included in the discussion section about representation of physical processes involved in moisture transport in East Asia in the reanalysis and MetUM simulations. You could briefly compare your findings with previous moisture source/transport diagnostic studies in East/Southeast Asia using Langragian models and reanalysis data (e.g. Sun and Wang, 2015; Baker et al. 2015; Chu et al. 2017).

The focus of this study is the comparison between reanalysis and simulations, as well as the sensitivity of simulated moisture sources to horizontal resolution and atmosphere–ocean coupling. We agree with the reviewer that understanding the physical processes that connect the moisture sources with the precipitation in target regions is equally important. In fact, we have prioritised the connection with the physical processes by publishing results on this topic prior to evaluating simulations. Details can be found in Guo et al. (2018, 2019).

We will add a comparison with the previous studies to the Discussion on Page 10–11 from Lines 25 onward: "*Moisture sources tracked using the WAM-2layers and the physical processes that link the source regions with the precipitation over EA have been discussed in detail in Guo et al. (2019). Compared with studies employing other moisture methods, the results are consistent (Sun and Wang, 2015; Baker et al., 2015; Chu et al., 2017). As also shown herein, the Indian Ocean provides the largest portion of moisture during boreal summer for precipitation over southeast EA. This contribution to precipitation decreases with the latitude of precipitation. Meanwhile, the moisture contribution from land sources increases with latitude. Local evaporation makes a larger contribution over the Tibetan Plateau compared to other EA subregions. During the boreal winter, due to the prevailing westerly and the frozen soil over the Eurasian continent, the Mediterranean Sea and other adjacent waterbodies become the major moisture contributor for precipitation over the mid–latitude EA subregions. MetUM simulations can generally capture most of these contributions, albeit with notable biases that vary with resolution and coupling. Similar biases have also been reported in Peatman and Klingman (2018), Stephan et al. (2017a, b).*"

– Although ERA–Interim shows indeed a good skill in simulating mean and inter–annual variations in land precipitation over East Asia this is not always the case for water cycling over the ocean. For example, P–E interannual variability in the tropical Indian Ocean is not well represented in ERA–Interim as compared to observationally–based products (see Skliris et al. 2014). This may affect the simulation of moisture transport from Indian Ocean moisture sources for SE Asia precipitation in ERA–Interim. In general, there are large discrepancies

between the different reanalyses in representing E & P variations over the ocean (see Schanze et al. 2010). A more critical discussion is needed in the text concerning the use of a single reanalysis product as a benchmark to compare moisture sources traced from climate model simulations.

We thank the reviewer for this suggestion. We will add a discussion on Page 11 between Lines 3–10: "*ERA-Interim is employed here for evaluating the simulations. It is chosen for its small residual in the global hydrological budget, its accurate representation of the mean and interannual variability of EA monsoon precipitation and its resemblance to the observation of evaporation over China (Trenberth et al., 2011; Lin et al., 2014; Sun and Wang, 2015). However, the ERA-Interim has noticeable biases in the representation of the water cycle over the ocean, i.e., the P-E interannual variability in the tropical Indian Ocean is not well represented compared to observations (Skliris et al., 2014; Schanze et al., 2010). This bias could potentially affect the moisture contribution from the Indian Ocean estimated with ERA-Interim. To deliver more accurate information on the performance of MetUM in terms of tracking moisture sources, multiple reanalysis datasets should be included, so that biases from any single reanalysis dataset can be identified and considered.*".

– I would suggest to additionally use the ERA5 dataset (which replaced ERA-Interim a year ago) in your analysis which has much higher horizontal resolution (_30km) and with considerable improvements w.r.t. ERA-interim including better global balance of precipitation and evaporation and better precipitation over land, especially in the tropical regions. In addition, this way you may also investigate the impact of higher model resolution on the reanalysis biases and compare changes due to increasing resolution in products with similar resolution in ECMWF and MetUM products (i.e. ERA-Interim/AN216 vs. ERA5/AN512). Although I recognise that this requires a considerable extra effort and while the paper is publishable in its current form, I think it could strengthen your analysis and further improve the robustness of your findings.

We will include the ERA5 in a future multi-reanalysis comparison of moisture sources, which has been suggested by the reviewer in a previous comment. However, at this stage of the work, it is too much effort to recompute the moisture sources and model biases against ERA5, rather than the ERA-Interim. Although ERA5 is an improvement on ERA-Interim, there are few studies published so far to suggest a better representation of the circulation in ERA5 for East Asia and the surrounding regions. The purpose of the manuscript is to show the large-scale biases in moisture sources in MetUM, which we think are adequately depicted when MetUM is compared against ERA-Interim, which was the state-of-the-art reanalysis when we performed the analysis and the WAM-2layers simulations.

Minor points

– You should provide the ERA-Interim space grid resolution in section 2.1

ERA-Interim space grid resolution has been specified in Section 2.1.

– I would suggest to use MetUM AN216 (rather than AN96) to compare with ERA-Interim in figures 2 & 4 as these two datasets have similar horizontal grid resolution.

As answered in previous comment, we downloaded ERA-Interim on a 1.5°×1.5° grid from its data portal. Therefore, keeping the comparisons with AN96 in Figures 2 and 4 seems reasonable. As mentioned in our manuscript, the sensitivity of simulated moisture sources to horizontal resolution (i.e., the difference between AN96 and AN216) is small compared to the model bias of either simulation against ERA-Interim. Figures 2 and 4 look quantitatively similar when replacing AN96 with AN216:

[Figure]

Figure R1. Annual mean precipitation of (a) MetUM AN216 and its difference with APHRODITE. Units: mm/day. The annual precipitation are averaged over 1982-2012.

[Figure]

Figure R2. Annual mean moisture source for EA subregions (a, c, e, g and i, units: mm/month) and vertically integrated moisture flux (vector, units: m³/s) calculated from ERA−Interim. Moisture source accounts for 80% of precipitation is shown. Difference in annual mean

moisture sources between AN216 and ERA-Interim (b, d, f, h and j). Units: mm/month (Skliris et al. 2014). Black box in each panel indicates the target region.

- Table 1: Please indicate units for horizontal grid resolution (degrees) and vertical resolution (levels)

Units have been added to the Table 1. Text in both the Table 1 caption and Section 2.2 have been modified.

- Typing errors Line 274: " . . . the eastern Tibetan Plateau, where the sruface is wetter. . ." Change into "surface

Correction has been made.

**References**

Baker, Alexander J., Harald Sodemann, James U. L. Baldini, Sebastian F. M. Breitenbach, Kathleen R. Johnson, Jeroen Van Hunen, and Pingzhong Zhang. 2015. 'Seasonality of westerly moisture transport in the East Asian summer monsoon and its implications for interpreting precipitation δ18O', *Journal of Geophysical Research: Atmospheres*, 120: 5850-62.

Chu, Qu-Cheng, Qi-Guang Wang, and Guo-Lin Feng. 2017. 'Determination of the major moisture sources of cumulative effect of torrential rain events during the preflood season over South China using a Lagrangian particle model', *Journal of Geophysical Research: Atmospheres*, 122: 8369-82.

Guo, Liang, Nicholas P. Klingaman, Marie-Estelle Demory, Pier Luigi Vidale, Andrew G. Turner, and Claudia C. Stephan. 2018. 'The contributions of local and remote atmospheric moisture fluxes to East Asian precipitation and its variability', *Climate Dynamics*.

Guo, Liang, Ruud J. Van Der Ent, Nicholas P. Klingaman, Marie-Estelle Demory, Pier Luigi Vidale, Andrew G. Turner, Claudia C. Stephan, and Amulya Chevuturi. 2019. 'Moisture Sources for East Asian Precipitation: Mean Seasonal Cycle and Interannual Variability', *Journal of Hydrometeorology*, 20: 657-72.

Skliris, Nikolaos, Robert Marsh, Simon A. Josey, Simon A. Good, Chunlei Liu, and Richard P. Allan. 2014. 'Salinity changes in the World Ocean since 1950 in relation to changing surface freshwater fluxes', *Climate Dynamics*, 43: 709-36.

Sun, Bo, and Huijun Wang. 2015. 'Analysis of the major atmospheric moisture sources affecting three sub-regions of East China', *International Journal of Climatology*, 35: 2243-57.